# Tailoring T$_{fh}$ profiles enhances antibody persistence to a clade C HIV-1 vaccine in rhesus macaques

Anil Verma[1†], Chase E Hawes[2,3†], Sonny R Elizaldi[2,4], Justin C Smith[5], Dhivyaa Rajasundaram[6], Gabriel Kristian Pedersen[7], Xiaoying Shen[8,9,10], LaTonya D Williams[8,9,10], Georgia D Tomaras[8,9,10,11,12], Pamela A Kozlowski[5], Rama R Amara[13,14], Smita S Iyer[1,3,4]*

[1]Department of Pathology, School of Medicine, University of Pittsburgh, Pittsburgh, United States; [2]Graduate Group in Immunology, University of California, Davis, Davis, United States; [3]California National Primate Research Center, University of California, Davis, Davis, United States; [4]Department of Pathology, Microbiology, and Immunology, School of Veterinary Medicine, University of California, Davis, Davis, United States; [5]Department of Microbiology, Immunology, and Parasitology, Louisiana State University Health Sciences Center, New Orleans, United States; [6]Bioinformatics Core, Department of Pediatrics, UPMC Children's Hospital of Pittsburgh, Pittsburgh, United States; [7]Statens Serum Institute, Copenhagen, Denmark; [8]Center for Human Systems Immunology, Durham, United States; [9]Department of Surgery, Duke University Medical Center, Durham, United States; [10]Duke Human Vaccine Institute, Duke University Medical Center, Durham, United States; [11]Department of Molecular Genetics and Microbiology, Duke University Medical Center, Durham, United States; [12]Department of Integrative Immunobiology, Duke University Medical Center, Durham, United States; [13]Department of Microbiology and Immunology, Emory University, Atlanta, United States; [14]Yerkes National Primate Research Center, Emory University, Atlanta, United States

**\*For correspondence:**
siyer.3@pitt.edu

[†]These authors contributed equally to this work

**Competing interest:** The authors declare that no competing interests exist.

**Abstract** CD4 T follicular helper cells (T$_{fh}$) are essential for establishing serological memory and have distinct helper attributes that impact both the quantity and quality of the antibody response. Insights into T$_{fh}$ subsets that promote antibody persistence and functional capacity can critically inform vaccine design. Based on the T$_{fh}$ profiles evoked by the live attenuated measles virus vaccine, renowned for its ability to establish durable humoral immunity, we investigated the potential of a T$_{fh}$1/17 recall response during the boost phase to enhance persistence of HIV-1 Envelope (Env) antibodies in rhesus macaques. Using a DNA-prime encoding gp160 antigen and T$_{fh}$ polarizing cytokines (interferon protein-10 (IP-10) and interleukin-6 (IL-6)), followed by a gp140 protein boost formulated in a cationic liposome-based adjuvant (CAF01), we successfully generated germinal center (GC) T$_{fh}$1/17 cells. In contrast, a similar DNA-prime (including IP-10) followed by gp140 formulated with monophosphoryl lipid A (MPLA) +QS-21 adjuvant predominantly induced GC T$_{fh}$1 cells. While the generation of GC T$_{fh}$1/17 cells with CAF01 and GC T$_{fh}$1 cells with MPLA +QS-21 induced comparable peak Env antibodies, the latter group demonstrated significantly greater antibody concentrations at week 8 after final immunization which persisted up to 30 weeks (gp140 IgG ng/ml- MPLA; 5500; CAF01, 2155; p<0.05). Notably, interferon $\gamma$+Env-specific T$_{fh}$ responses were consistently higher with gp140 in MPLA +QS-21 and positively correlated with Env antibody persistence. These findings suggest that vaccine platforms maximizing GC T$_{fh}$1 induction promote persistent Env antibodies, important for protective immunity against HIV.

## eLife assessment

The authors' findings have theoretical or practical deep implications, which makes them **important**. The methods, data, and analyzes support the authors' arguments with only minor weaknesses, and overall they are **solid**. In vitro culture experiments could provide evidence to strengthen the evidence for the functional significance of Th1-mediated cytokines in the observed B cell responses.

## Introduction

Despite options for testing and treatment, progress against HIV is slowing, emphasizing the urgent need for an HIV vaccine (*Fauci, 2017*; *Eisinger and Fauci, 2018*). While advances have been made in immunogen design and vaccine approaches, the main challenge lies in generating durable, high-affinity antibodies against the HIV-1 Envelope (Env) glycoprotein, crucial for protection. To design a vaccine that confers long-term protective immunity, effective stimulation of CD4 T follicular helper ($T_{fh}$) cells is essential, as they provide vital costimulatory and cytokine support to B cells within germinal centers (GC), leading to the production of persistent antibodies following immunization (*Slifka et al., 1998*; *Crotty, 2014*). $T_{fh}$ cells possess distinctive $T_h1$, $T_h2$, and $T_h17$-type cell attributes, programmed by the inflammatory response during the T-cell priming phase, with each $T_{fh}$ subset differentially contributing to GC B cell proliferation, survival, and differentiation (*Morita et al., 2011*; *Barbet et al., 2018*; *Gao et al., 2023*). Therefore, in addition to antigen selection, promoting the differentiation and expansion of optimal $T_{fh}$ subsets to generate potent and enduring humoral immunity is critical for vaccine design.

Clinical studies have demonstrated that booster immunization with a $T_h1$ glucopyranosyl lipid adjuvant-stable emulsion (GLA-SE)-formulated malaria antigen leads to enhanced antibodies at memory time points compared to an aluminum adjuvanted vaccine (*Hill et al., 2019*). We demonstrated that promoting the induction of $T_{fh}1$ cells by utilizing interferon-induced protein (IP)10, a ligand for and an inducer of CXCR3, as a molecular adjuvant to a DNA vaccine ($DNA_{IP10}$) followed by boosting with protein adjuvanted with Army Liposome Formulation (ALFQ) consisting of liposomal monophosphoryl lipid A (MPLA) plus a saponin derivative, QS-21 (*Alving et al., 2012*; *Rao et al., 2018*) enhanced GC responses, increased HIV anti-Env binding antibodies, and stimulated significantly higher cross-clade reactivity with increased specificity to V1V2 conformational epitopes, and higher avidity (*Verma et al., 2019*). Similarly, a recent macaque study utilizing Clade C DNA + GLA SE adjuvanted Env protein immunization revealed that frequencies of Env-specific CD28 + IFNγ+ cells, indicative of $T_h1$ responses, correlated with the development of Env antibodies following the final immunization (*Felber et al., 2020*).

In addition to IFNγ, interleukin 17 (IL-17) has been shown to enhance cognate T-B cell interactions, leading to an effective GC response (*Moyron-Quiroz et al., 2004*; *Mitsdoerffer et al., 2010*; *Kumar et al., 2013*). Studies conducted in mice have further demonstrated that $T_{fh}17$ cells support efficient antibody recall responses (*Gao et al., 2023*). Notably, the persistence of circulating $T_{fh}17$ cells specific to measles has been observed in adults who have received the live attenuated measles virus vaccine (LAMV) during childhood, suggesting that measles vaccine triggers $T_{fh}17$ responses which persist beyond the GC phase (*Gao et al., 2023*). Intriguingly, earlier studies exploring peripheral CD4 T cell determinants of serological memory elicited by LAMV have led to the hypothesis that LAMV promotes both $T_h1$ and $T_h2$ CD4 differentiation programs, based on production of IFNγ+ and IL-4, which are cytokines also produced by $T_{fh}$ cells (*Ovsyannikova et al., 2003a*; *Ward and Griffin, 1993*; *Ovsyannikova et al., 2003b*). However, our understanding of the GC $T_{fh}$ cell response to LAMV, which plays a crucial role in establishing long-term immunity against measles, remains incomplete.

In this study, based on GC $T_{fh}$ profiles evoked by LAMV, we explored the potential of a $T_{fh}1$ or mixed $T_{fh}1/17$ targeted vaccine to enhance HIV-1 Env antibodies in rhesus macaques using an DNA-prime/protein boost approach. After booster immunization with a Clade C gp140 protein formulated in CAF01 (*Wørzner et al., 2021*), a cationic liposome-based formulation, we observed the generation of $T_{fh}1/17$ cells within GCs, while $T_{fh}1$ cells were induced when MPLA +QS-21 was used as the adjuvant. Notably, stimulating $T_{fh}1$ cell induction with MPLA resulted in significantly greater antibody persistence at week 8 and up to 30 weeks after the final immunization. Moreover, the $T_{fh}1$ regimen with MPLA adjuvant led to higher tier 1 neutralization titers, increased levels of IgG1 subclass antibodies, and improved antibody effector functions. These findings highlight a unique potential of $T_{fh}1$

cells in promoting humoral immunity against HIV and suggest that vaccine strategies maximizing $T_{fh}1$ cell induction may hold promise for eliciting durable protective immunity against HIV.

## Results

### GC $T_{fh}1$ and GC $T_{fh}17$ cells recalled by measles booster

To decipher the specific $T_h$ subset of $T_{fh}$ cells contributing to the establishment of durable serological memory, we employed live attenuated measles virus vaccine (LAMV) as a model and investigated $T_{fh}$ responses in peripheral blood and lymph nodes (LN) in a cohort of 16 healthy adult female rhesus macaques (*Figure 1A*). Consistent with documented evidence of long-term persistence of the memory B cell response to LAMV, rapid recall of measles virus (MeV) IgG ensued at week 2 post LAMV, irrespective of the booster interval (*Figure 1B*). By week 20, MeV IgG concentrations displayed a 2.6-fold increase compared to baseline, signifying the efficacy of LAMV booster in generating persistent antibodies. These augmented levels of memory antibodies were accompanied by enhanced antibody avidity (*Figure 1C*), underscoring robustness of the humoral immune response elicited by LAMV.

Tracking MeV-specific $T_{fh}$ cells using the activation-induced marker (AIM) assay showed induction of circulating (c)$T_{fh}$ cells at day 7 post-vaccination (*Figure 1—figure supplement 1A*). Confirming $T_{fh}$ cell induction in response to LAMV, activated cTfh cells, identified by co-expression of ICOS and Ki-67, were transiently upregulated at day 7 (*Figure 1D*, *Figure 1—figure supplement 1B*). These responding cTfh cells were PD-1 +and their frequencies at day 7, but not day 0, significantly correlated with MeV IgG at Week 20 (*Figure 1E*). We next proceeded to analyze heterogeneity within cTfh cells with respect to expression of $T_h1/2/17$ chemokine receptors. Our temporal analysis of the responding cTfh compartment demonstrated significant induction of CXCR3 +$T_{fh}1$ cells at day 7 (*Figure 1F*). In contrast, we observed a decrease in frequencies of CCR4-expressing cTfh2 subsets and CCR6-expressing cTfh17 subsets. Utilizing Boolean analysis, we further identified that the majority of responding cTfh cells expressed CXCR3, indicating induction of cTfh1 cells in response to LAMV (*Figure 1G*).

Expanding our investigation to include phenotypic and molecular features of CXCR5$^+$ PD-1$^{++}$ GC $T_{fh}$ cells (*Figure 1H*, *Figure 1—figure supplement 1C*), we found that of the GC $T_{fh}$ cells expressing either CXCR3 or CCR6, CXCR3 expression predominated (median %CXCR3+, 28%). Additionally, we observed that on average 7% of GC $T_{fh}$ cells expressed CCR6. Notably, our analysis did not reveal substantial expression of CCR4 within GC $T_{fh}$ cells (*Figure 1I*). Molecular analysis of sorted CXCR5 +PD-1+/++LN $T_{fh}$ cells (*Figure 1—figure supplement 1D*) further provided insights into their functional specialization, with the induction of key transcription factors BATF and IRF4, as well as the expression of chemokine receptors CCR5 and CCR6, indicating the activation of $T_h1$ and $T_h17$ programs within GC $T_{fh}$ cells (*Figure 1J–K*). Collectively, these findings demonstrate that both blood and GC $T_{fh}$ cells, during the peak effector response following LAMV immunization, exhibit a predominant $T_{fh}1$ helper bias, with GC $T_{fh}$ cells also demonstrating a $T_{fh}17$ bias.

### HIV-1 vaccine modalities for inducing GC $T_{fh}1$ and GC $T_{fh}17$ subsets

Building on our observations made with LAMV, we next investigated the potential of fine-tuning $T_{fh}$ responses towards $T_{fh}1$ and $T_{fh}17$ profiles for generation of persistent anti-HIV-1 Env antibodies. We immunized two cohorts of rhesus macaques against HIV Env using a DNA prime and protein boost vaccination strategy tailored towards either a $T_{fh}1$ (n=6) or mixed $T_{fh}1/17$ (n=6) response (*Figure 2—figure supplement 2*). For both vaccine regimens, animals received an initial intradermal DNA prime vaccine encoding HIV-1 Env gp160 (C.1086C) at weeks 0, 4, 8, followed by an subcutaneous HIV C.ZA 1197 MB Env gp140 protein boost vaccination at weeks 12 and 20. To tailor vaccine induced $T_{fh}$ responses, we included additional $T_{fh}$ polarizing factors with DNA prime vaccines encoding for cytokines IP-10 ($T_{fh}1$ regimen) or IP-10 and IL-6 (mixed $T_{fh}1/17$ regimen), coupled with corresponding gp140 protein boost formulated in either the $T_h1$ polarizing adjuvants, MPLA +QS-21, or the $T_h1/17$ polarizing CAF01 adjuvant platform, respectively. Over the course of both the priming and boost stages, we performed routine collection of both blood and rectal secretions from animals to assess the extent of systemic and mucosal humoral immunity elicited by the two regimens. Additionally, we collected inguinal LN biopsies and fine needle aspirates of LN (FNA) at baseline, during the DNA prime stage, and following both protein boost to comprehensively characterize vaccine-elicited GC

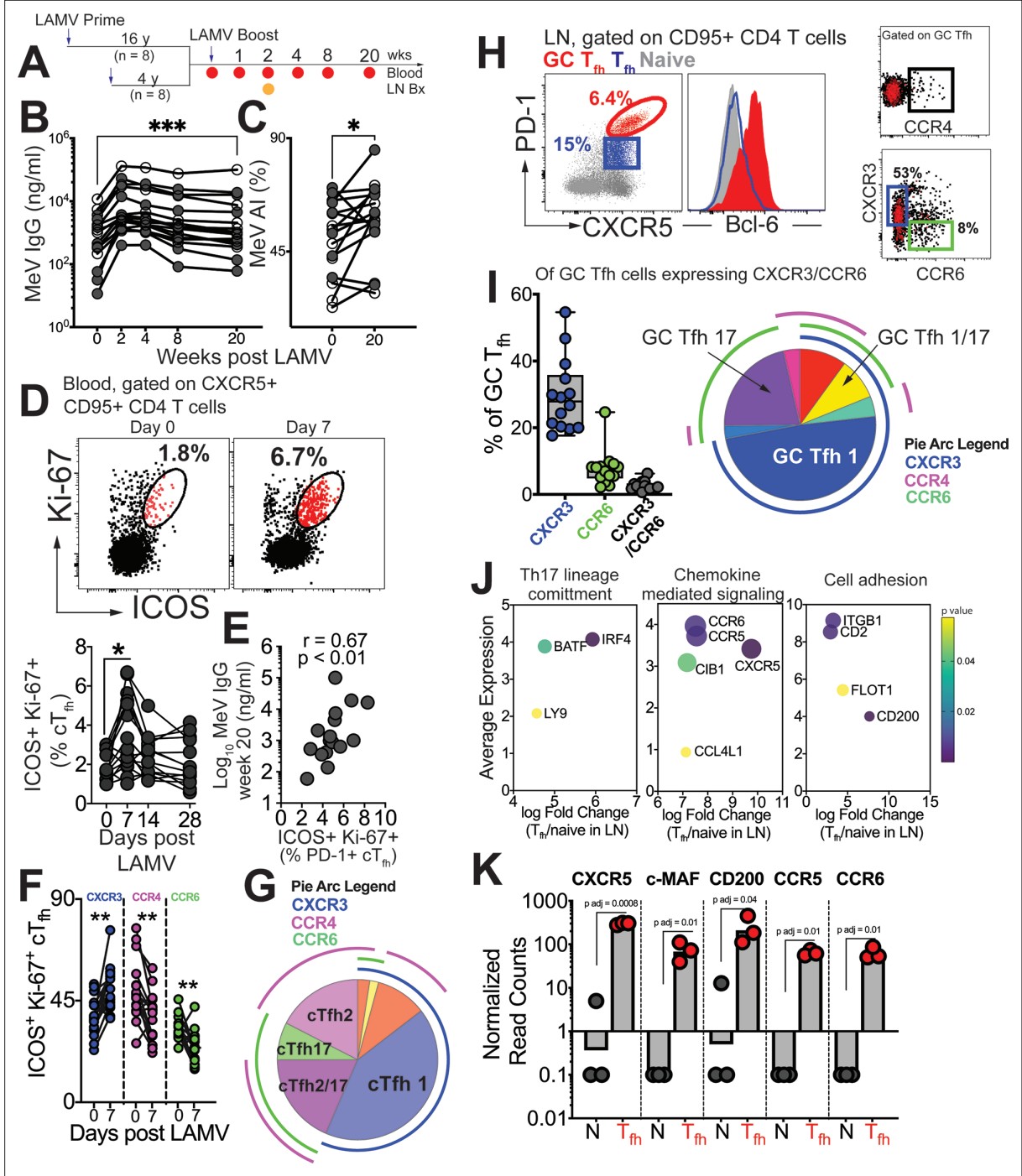

**Figure 1.** GC T$_{fh}$1 and GC T$_{fh}$17 cells recalled by measles booster. (**A**) Overview of study design; adult female rhesus macaques immunized with Live Attenuated Measles Virus vaccine (LAMV). (**B**) Serum MeV IgG kinetics measured by ELISA (filled circles, aged rhesus; open circles, young adults). (**C**) Avidity index (AI) of MeV IgG measured using chaotropic displacement ELISA with sodium thiocyanate at week 20 and week 0 in serum. (**D**) Representative flow cytometry plots showing ICOS$^+$Ki-67$^+$ circulating (c)Tfh cells in blood at day 0 and day 7 post LAMV. Kinetics of ICOS$^+$Ki-67$^+$ cTfh cells. (**E**) Correlation between ICOS + Ki-67+cTfh cells at day 7 and MeV IgG at week 20. (**F**) T$_h$ profile of cTfh cells shows induction of CXCR3$^+$ cTfh1 at day 7. (**G**) Boolean analysis (n=16) shows cTfh1 cells induced 1 week post LAMV. Overlapping pie arcs denote cTfh cells expressing multiple chemokine receptors as denoted by arc color. (**H**) Representative flow cytometry plots show CXCR5$^+$ PD-1$^{++}$ GC T$_{fh}$ cells and histogram shows Bcl-6 expression on GC T$_{fh}$ cells. T$_h$ profile of GC T$_{fh}$ cells shows expression of CXCR3 and CCR6. (**I**) Boolean analysis of GC T$_{fh}$ cells expressing either CXCR3 or CCR6 (n=14) shows proportion of T$_h$1, T$_h$17 and T$_h$1/17 GC T$_{fh}$ subsets. (**J**) Bubble plots show genes for significantly enriched pathways related to T helper differentiation on sorted CXCR5 + PD-1+/++ cells. (**K**) Gene expression on sorted CXCR5 +PD-1+/++ cells. Statistical analysis was performed using two-tailed Wilcoxon matched-pairs signed rank test (in panels B-D, F) or spearman rank correlation test (**E**); * p<0.05, **p<0.01, *** p<0.001, **** p<0.0001.

*Figure 1 continued on next page*

*Figure 1 continued*

The online version of this article includes the following figure supplement(s) for figure 1:

**Figure supplement 1.** MeV specific Tfh cells.

T$_{fh}$ responses in draining LNs. GC responses typically peak between 2 and 3 weeks post first booster with more rapid recall kinetics after the second booster (*Iyer et al., 2015*). Our rationale for sampling LN 3 weeks following the second booster was to assess whether the reported effects of CAF01 on sustained antigen release might result in GC persistence (*Henriksen-Lacey et al., 2011*; *Pedersen et al., 2020*).

## Induction of robust T cell activation in GCs with HIV-1 Env formulated in CAF01 and MPLA+QS-21 adjuvants

To gain insights into LN T$_{fh}$ responses induced by MPLA and CAF01 platforms, we utilized complementary approaches of in-situ protein expression analysis of GCs and RNA sequencing of Env-stimulated CD95 +CD4 T cells of selected animals in each vaccine group (*Figure 2A*). To track immunological changes within GCs, we applied the GeoMx Digital Spatial Profiler to assess protein expression of key immune targets longitudinally at baseline (BL, prior to DNA immunization), week 2 post-protein 1 (P1w2) and week 3 post-protein 2 (P2w3) immunizations. We stained tissue sections with morphological markers (CD20, CD3, Ki-67) and collected circular regions of interest (ROI) based on co-localization of CD3 with CD20 +Ki-67+GC B cells. For each tissue, two to three ROIs were selected to ensure adequate representation of GCs within each animal. This approach allowed us to accurately identify GCs and analyze protein expression of both T and B cells within these specific regions (*Figure 2B*).

Thirty-two protein targets were profiled simultaneously, along with isotype controls (Ms IgG1, Ms IgG2A, Rb IgG) and housekeeping proteins (S6, Histone H3, GAPDH). After applying quality control measures, we calculated the signal-to-noise ratio for each target which provided robust assessment of protein expression levels (). CD45 and B2M were among the most abundantly expressed proteins while LAG3 and PD-L1 demonstrated low signal expression. To address variations in ROI surface area and tissue quality, normalized protein expression (NPE) was calculated for each protein. NPE analysis revealed robust expression of key lineage proteins, CD45, CD20, CD3, CD4, and CD8, as well as GC markers PD-1 and Ki-67 within ROIs (*Figure 2C*). We observed notable associations between protein expression patterns within the GC microenvironment; specifically, a strong correlation between the expression of CD4 and CD20 was observed (). Furthermore, a significant association between the co-stimulatory molecule 4-1BB and CD4 expression () was observed within the GC. Association of the innate immune protein stimulator of interferon genes (STING) with both 4-1BB and CD4 expression suggested a potential interplay between innate and adaptive immune pathways within the context of GC responses. Altogether, the protein expression data provide evidence for coordinated expression of key proteins involved in GC function in both vaccine groups ().

To identify specific proteins and pathways activated in response to vaccination, we performed differential expression analysis on a per-protein basis. We modeled NPE using a linear mixed-effect model (LMM), which allowed us to account for sampling multiple ROIs per tissue within each animal (FDR p value cut-off,<0.05). Volcano plots illustrate proteins induced at P1w2 (*Figure 2D*) and P2w3 (*Figure 2E*) following vaccination. Induction of 4-1BB, ICOS, PD-1, CD44 post boost indicated activation of T cells to antigen stimulation within GCs in both vaccine groups (*Figure 2F*). The higher relative expression of these proteins at P2w3 was consistent with the reported effects of CAF01 in GC persistence. We found that expression of the checkpoint inhibitors CTLA4 and PD-L1 was significantly reduced post CAF01 vaccination, while CD4 expression increased. In contrast, expression of the regulatory enzyme, IDO1; checkpoint inhibitor, PD-L2; and CD256 (B7-H3), and the innate sensor, STING, was significantly higher post vaccination in the MPLA group (*Figure 2D–E and G*).

Across vaccine regimens, significant differences were observed at P1w2 (*Figure 2H*). Significantly higher expression of Ki-67 with MPLA compared to CAF01 was suggestive of a more rapid GC response (*Figure 2H*). Furthermore, the checkpoint inhibitory receptors LAG3, CTLA4, as well as IDO1, exhibited higher expression levels with MPLA at P1w2. Importantly, no significant differences were observed at BL, validating the specificity of immune activation to vaccination. Altogether, the in situ protein analysis indicated robust induction of GC responses across vaccine platforms with distinct

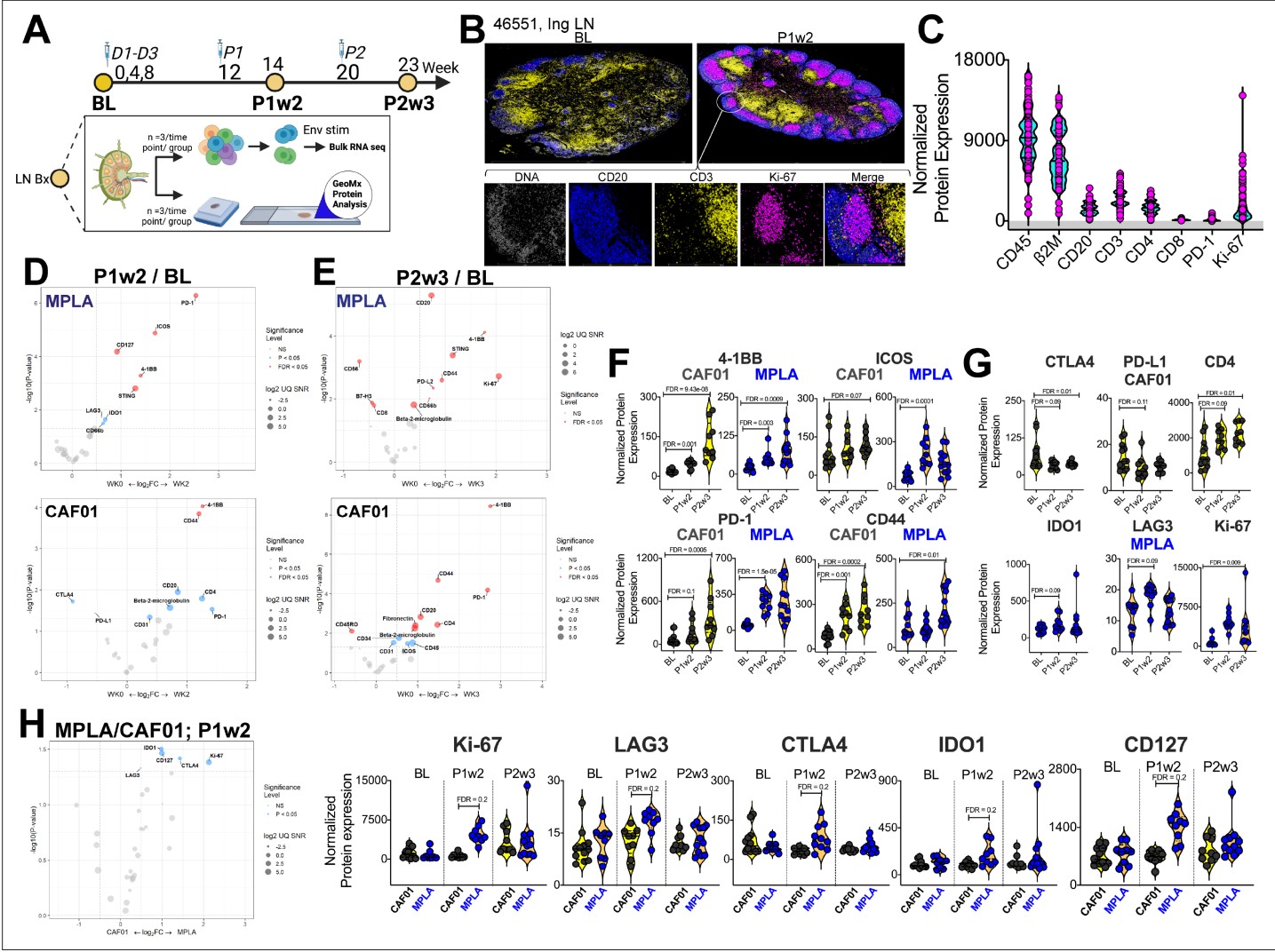

**Figure 2.** Induction of Robust T Cell Activation in GCs with HIV-1 Env Formulated in CAF01 and MPLA +QS-21 Adjuvants. (**A**) Overview of experimental design for in-situ proteomics and transcriptional analysis. (**B**) Representative images of GC from FFPE sections of inguinal LN at baseline and week 2 post protein 1 immunization with HIV-Env gp140 | MPLA +QS-21; scale bar; top, 3 mm, close up, 300 µm. LN sections were stained for CD20 (blue), CD3 (yellow), Ki-67 (magenta), and DNA (gray) to identify GCs. Circular ROIs (100 µm in diameter, total 60) were selected based on co-localization of CD3 with CD20 +Ki-67+GC B cells for proteomic profiling with a 32-plex antibody cocktail. Normalized protein expression (NPE) values were calculated using three negative control IgG probes. (**C**) NPE of key lineage markers across all ROIs at BL, P1w2, and P2w3. (**D–E**) Volcano plots show proteins induced post boost in each vaccine group. (**F**) Violin plots show common proteins induced post protein boost in MPLA and CAF01 groups. (**G**) Violin plots of proteins induced with CAF01 and MPLA. (**H**) Volcano plot (left) and Violin plots (right) of proteins significantly different across HIV-1 Env gp140 MPLA +QS-21 and CAF01 regimens. Differential expression was modeled using a linear mixed-effect model to account for the sampling of multiple ROI/AOI segments per patient/tissue. Criterion of significance was nominal p-value <0.05, plots show False discovery rate (FDR).

The online version of this article includes the following figure supplement(s) for figure 2:

**Figure supplement 1.** Experimental design of DNA-prime/Protein-boost Clade C HIV-1 immunization.

**Figure supplement 2.** Sequencing QC histograms show (**A**) Signal-to-noise ratio computed by dividing raw count values by the geometric mean of the negative IgG probes.

qualitative and quantitative effects initiated by MPLA versus CAF01 suggesting potential differences in regulatory mechanisms and immune activation pathways between the vaccine regimens.

### T$_h$1 molecular programs potentiated by HIV-1 Env/MPLA+QS-21

To assess the extent of T$_h$1/T$_h$17 programming in CD4 T cells induced by vaccination, we profiled the transcriptome of CD95 +CD4 T cells isolated from the LN. We sort purified and RNA sequenced

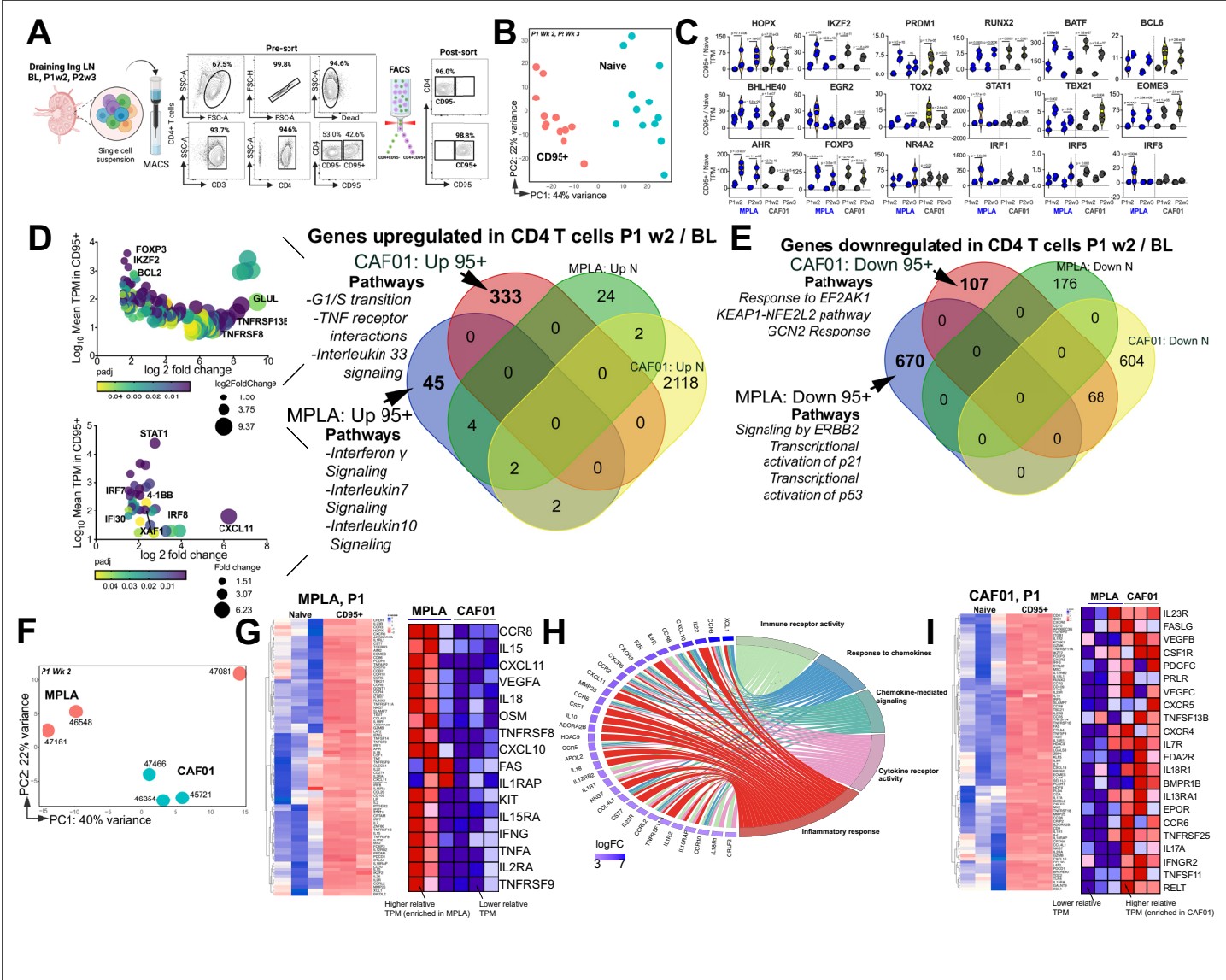

**Figure 3.** T$_h$1 molecular programs potentiated by HIV-1 Env/MPLA + QS-21. (**A**) FACS purified CD95- (naive [N]) and CD95 + CD4 T cells, stimulated overnight with HIV-1 Env, were sequenced to assess transcriptional programs. (**B**) Principal component analysis (PCA) of all expressed genes shows distinct clustering of N and CD95 + cells. (**C**) Violin plots show transcripts per million (TPM) values of differentially expressed transcription factors (p adj <0.05) in CD95 + CD4 T cells relative to naive CD4 T cells post vaccination. (**D**) Bubble plots depict DEG at P1w2 versus BL in CD95 + CD4 T cells in CAF01 (top) and MPLA (bottom). Venn Diagram of DEG genes upregulated in CD95 + and naive subsets at P1w2 relative to BL shows 45 and 333 genes exclusively upregulated in CD95 + CD4 T cells with MPLA and CAF01, respectively. (**E**) Venn diagram of significantly downregulated genes. (**F**) PCA of all expressed genes shows distinct clustering across vaccine groups at P1w2. (**G**) Heat maps depict log$_2$ gene expression (transcripts per million (TPM)) for highly DEG in CD95 + cells compared to naive cells at P1w2 in MPLA; heat map in inset depicts TPM of genes represented in Cytokine-cytokine receptor interaction pathways across vaccine regimens. (**H**) Chord plot shows GO Terms enriched with corresponding upregulated genes in CD95 + CD4 T cells at P1w2 with MPLA. (**I**) Heat maps depict log$_2$ gene expression (TPM) for highly DEG in CD95 + cells compared to naive cells at P1w2 in CAF01; heat map in inset depicts TPM of genes represented in Cytokine-cytokine receptor interaction pathway across vaccine regimens.

The online version of this article includes the following figure supplement(s) for figure 3:

**Figure supplement 1.** QC Metrics of RNA sequencing.

naive (CD95-) and CD95 + CD4 T cells at BL, P1w2, and P2w3 following overnight stimulation with HIV-1 Clade C Env peptide pools. A total of 18 samples per group, which included three biological replicates per subset, were sequenced in parallel resulting in over 25 million high-quality reads per sample (*Figure 3A*, *Figure 3—figure supplement 1A-B*). The transcriptomes of naive versus CD95 + CD4 T cells post immunization, in alignment with their distinctive biological states, exhibited

clear demarcations along distinct dimensions within the principal component analysis (PCA) plot (*Figure 3B*). Transcription factors (TF) involved in regulating distinct cellular differentiation programs were enriched in CD95 + CD4 T cells at P1w2 and P2w3. These included HOPX, IKZF2, PRDM1, RUNX2, associated with effector differentiation. Additionally, TF such as BATF, BCL6, BHLHE40, EGR2, and TOX2 involved in T$_{fh}$ differentiation were also identified. Furthermore, we identified genes regulating T$_h$1 (STAT1, TBX21, EOMES), T$_h$17 (AHR), and Treg (FOXP3, NR4A2) programs. Moreover, the expression of interferon regulatory factors (IRFs) controlling T cell differentiation (IRF1, IRF5, IRF8) was observed, demonstrating that CD95 + cells post vaccination encompassed effector and fully differentiated CD4 T cell subsets (*Figure 3C*).

We first assessed temporal changes in gene expression by focusing on differentially expressed genes (DEGs) within CD95 + cells at P1w2 and P2w3, compared to BL (Venn diagram in 3D). Notably, a significantly higher level of transcriptional perturbation was observed in response to CAF01, particularly at P2w3, within both naive and CD95 + CD4 T cells (top bubble plot in 3D, *Figure 3—figure supplement 1C*), suggesting that CAF01 induces extensive gene expression changes in a cell-extrinsic manner. By specifically examining genes induced exclusively in CD95 + cells, but not naive cells, at P1w2, we made several observations. The T$_h$1 transcriptional regulator STAT1, as well as IFNγ-induced genes such as IRF7, IRF8, IFI30, XAF-1, and CXCL11, were among the most highly expressed genes in MPLA (bottom bubble plot, 3D), highlighting the activation of T$_h$1 programs following vaccination. Indeed, interrogation of top 3 enriched pathways utilizing the Reactome database (*Fabregat et al., 2017*) revealed that genes encompassing IFNγ, IL-7, and IL-10 signaling networks were induced with MPLA at P1w2. In the CAF01 group, we observed enrichment of co-stimulatory molecules TNFRSF8 (CD30) and TNFRS13B (TACI), expressed by activated T cells, along with expression of the anti-apoptotic molecule BCL-2 and the transcription factor Foxp3.

Analysis of genes downregulated showed pathways regulating cell cycle inhibitors p21 and p53 were enriched with MPLA while pathways regulating cellular responses to stimuli and stress were downregulated with CAF01 (*Figure 3E*). Collectively, transcriptional profiles demonstrated activation of effector programs within CD4 T cells with common and unique pathways induced across vaccine platforms.

Consistent with this, CD95$^+$ CD4 T cells at P1w2 and P2w3 exhibited notable clustering patterns across vaccine regimens. Particularly, tighter, and more distinct clusters were observed at P1w2 compared to P2w3 notably with CAF01 (*Figure 3F*, *Figure 3—figure supplement 1D-E*). Exploration of genes regulating T$_h$1 and T$_h$17 programs within CD95 + CD4 T cells at P1w2 showed that CD95 + CD4 T cells in both MPLA and CAF01 groups exhibited overlapping expression profiles, with genes regulating major classes of cytokine-cytokine receptor interactions represented. The MPLA group exhibited higher expression levels of IFN-induced chemokines CXCL10 and CXCL11, IL1 family members IL18 and IL1RAP, IL15 family members IL15 and IL15RA, as well as TNF cytokine family members FAS, TNFα, TNFRSF8 (CD30), and TNFRSF9 (4-1BB) (*Figure 3G–H*, *Figure 3—figure supplement 1F*).

In contrast, the CAF01 group demonstrated expression of the T$_h$17 receptor IL23R, along with TNF family members FASLG, TNFRSF13B (TACI), TNFRSF11 (RANKL), and TNFRSF12 (DR3) within CD95 + cells with higher relative expression of IL17A observed with CAF01, indicative of T$_h$17 responses (*Figure 3I*, *Figure 3—figure supplement 1G*). Collectively, this gene-level analysis indicated strong engagement of T$_h$1 molecular programs within LN CD4 T cells with MPLA.

## Phenotypically and functionally specialized GC T$_{fh}$1 / T$_{fh}$17 subsets elicited with HIV-1 Env formulated in CAF01 and MPLA+QS-21 adjuvants

After confirming the robust induction of GC responses by both vaccines and observing distinct molecular programs in CD4 T cells, we proceeded to investigate the phenotypic and functional characteristics of T$_{fh}$ cells. We examined the GC T$_{fh}$ cell subset characterized by the expression of CXCR5 and high levels of PD-1, using the gating strategy outlined in *Figure 4—figure supplement 1A*. Our analysis of the lymph nodes revealed an increase in the frequencies of GC T$_{fh}$ cells following each protein boost in both groups compared to baseline. Importantly, there were no statistically significant differences observed between the two adjuvant platforms, as shown in *Figure 4A*. We further examined the frequencies of T$_{fh}$ cells characterized by the expression of CXCR5 and PD-1 (CXCR5 +PD-1+),

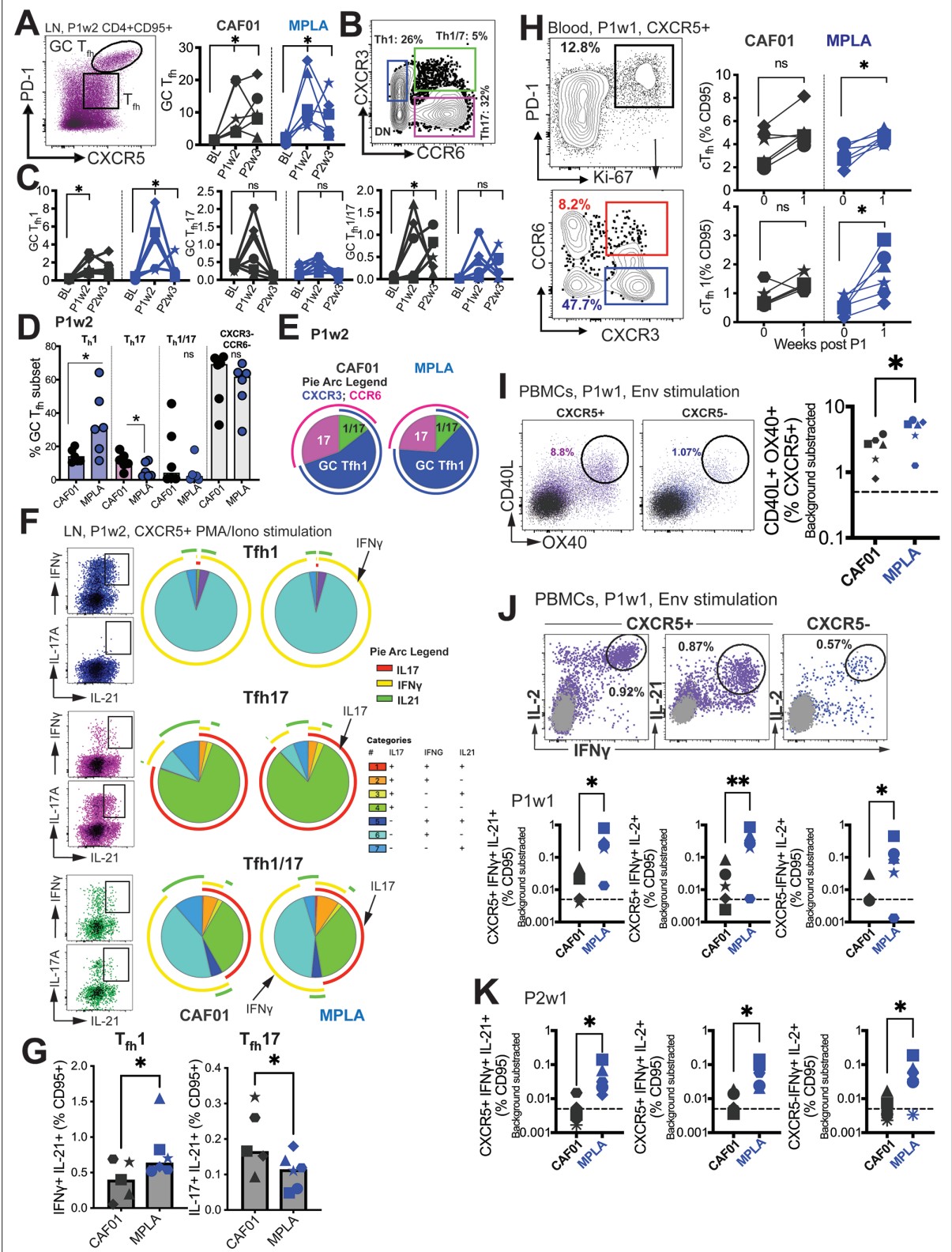

**Figure 4.** Phenotypically and functionally specialized GC T$_{fh}$1 /T$_{fh}$17 subsets elicited with HIV-1 Env Formulated in CAF01 and MPLA +QS-21 Adjuvants. (**A**) Representative flow cytometry plot illustrates GC T$_{fh}$ cells and GC T$_{fh}$ frequencies (%CD95) show in kinetic plot to right. (**B**) Flow cytometry plot shows identification of GC T$_{fh}$1, T$_{fh}$17, and T$_{fh}$1/17 cells with (**C**) temporal kinetics expressed as %CD95+. Significance indicates differences in GC T$_{fh}$ subsets at P1 and P2 relative to baseline. (**D**) Proportion of T$_{fh}$1, T$_{fh}$17, T$_{fh}$1/7 and CXCR3-, CCR6- GC T$_{fh}$ cells across CAF01 and MPLA at P1w2. (**E**) Boolean analysis

*Figure 4 continued on next page*

*Figure 4 continued*

(n=6 each group) shows proportion of T$_h$1, T$_h$17 and T$_h$1/17 GC T$_{fh}$ subsets. (**F–G**) Intracellular cytokine staining (ICS) analysis of T$_h$1, T$_{fh}$17, T$_{fh}$1/17 subsets at P1w2. (**H**) Gating strategy (left) and frequencies (right) of circulating T$_{fh}$ (cT$_{fh}$) and cT$_{fh}$1 cells in whole blood at P1w0 and P1w1. (**I**) Gating strategy (left) and frequencies (right) of activated (CD40L$^+$OX40$^+$) cT$_{fh}$ (CXCR5$^+$) and non-cT$_{fh}$ CD4 T cells (CXCR5-) cells in PBMCs at P1w1. (**J**) ICS of cTfh and non-cTfh CD4 T cells following Env stimulation at P1w1 and P2w1. Data points show individual animals. Statistical analysis was performed using two-tailed Wilcoxon matched-pairs signed rank test (in panels **A**, **C**, **H**) or Mann-Whitney U test (in panels **D**; **G**, **I–K**); * p<0.05, **p<0.01.

The online version of this article includes the following figure supplement(s) for figure 4:

**Figure supplement 1.** Gating strategy to identify GC Tfh and Tfh cells in lymph nodes.

which also exhibited a significant increase following the first protein boost in both groups (***Figure 4— figure supplement 1B***).

To gain insights into the T$_{fh}$ helper profiles within the GC, we assessed the expression of CXCR3 (T$_{fh}$1) and CCR6 (T$_{fh}$17) on the GC T$_{fh}$ cells (flow plot in B). Both adjuvants induced GC T$_{fh}$1 cells, as evidenced by the expression of CXCR3. However, a selective increase in GC T$_{fh}$1/17 cells was observed with CAF01, while the frequencies of GC T$_{fh}$17 cells did not show a significant increase following immunization (***Figure 4C***).

At P1w2, our investigation into the relative frequencies of polarized GC T$_{fh}$ subsets revealed significant insights. We observed that most GC T$_{fh}$ cells retained a non-polarized state (with respect to expression of CXCR3 and CCR6), with no substantial distinctions noted between the CAF01 and MPLA groups. Notably, the T$_h$1 GC subset demonstrated a significant increase in the MPLA group, while the T$_h$17 subset exhibited a higher prevalence in the CAF01 group (***Figure 4D***). This observation was further supported by Boolean analysis, which revealed a greater proportion of the T$_h$17 subset associated with CAF01, and a higher proportion of the GC T$_{fh}$1 subset with MPLA at P1w2, as illustrated in ***Figure 4E***.

Following the observed induction of LN T$_{fh}$1 responses, we quantified the Env-specific CD4 T cell response using intracellular cytokine staining (ICS). The analysis revealed higher relative frequencies of Env-specific T$_{fh}$ cells and Env-specific CXCR5- CD4 T cells producing IFNγ with MPLA compared to CAF01 (***Figure 4—figure supplement 1C***). To obtain a more comprehensive understanding of the cytokine profile within the LN T$_{fh}$1 and T$_{fh}$17 subsets, we assessed cytokine production after PMA/ Ionomycin stimulation (***Figure 4F***). By utilizing the discrete expression patterns of CXCR3 and CCR6 on CXCR5 +CD4 T cells in the LN, in conjunction with the canonical cytokines IFNγ (T$_h$1), IL-17 (T$_h$17), and IL-21 (T$_{fh}$), we made several important observations. Firstly, we found that within the T$_{fh}$1 subset, IL-21 +IFNγ+co-producing cells were more abundant, and their levels were significantly increased with MPLA stimulation (***Figure 4G***). Conversely, within the T$_{fh}$17 subset, we observed that IL-21 +IL-17+co-producing cells were more abundant, and their levels were significantly higher in the CAF01 group. Notably, the T$_{fh}$1/17 subset demonstrated the highest degree of polyfunctionality, as evidenced by the production of multiple cytokines.

We further analyzed the cytokine profiles by gating IFNγ and IL-21 co-producers versus IL-21 single producers. This analysis revealed that both subsets, T$_{fh}$1 and T$_{fh}$17, produced IL-2 and TNFα. However, IL-17 production was predominantly observed in IFNγ-negative, IL-21 + cells (***Figure 4— figure supplement 1D***). To assess systemic levels of IL-21, we measured serum IL-21. We found a significant induction of IL-21 in both vaccine groups at day 7 post-immunization, with no observable differences observed across the adjuvant platforms (***Figure 4—figure supplement 1E***). This suggests that the induction of IL-21 is a common feature of both vaccine formulations. Overall, these findings suggest that MPLA preferentially induces the production of IL-21 + IFNγ+ cells within the T$_{fh}$1 subset, while CAF01 promotes the generation of IL-21 + IL-17+ cells within the T$_{fh}$17 subset. Furthermore, the T$_{fh}$1/17 subset exhibits the most diverse cytokine production profile among the subsets examined.

Next, we investigated whether the effects of MPLA versus CAF01 adjuvants were also evident in cTfh responses. We observed a significant increase in proliferating cTfh cells, with a predominant induction of cTfh1 cells, similar to the phenotype observed in the LN, when MPLA was used as the adjuvant (***Figure 4H***). In contrast, no evidence of a T$_{fh}$17 or T$_{fh}$1/17 bias in cTfh cells was observed with CAF01 (data not shown). To evaluate Env-specific T$_{fh}$ cell abundance, we utilized the AIM assay (***Figure 4—figure supplement 1F***). The analysis showed a higher relative induction of Env-specific T$_{fh}$ responses with MPLA at P1w1 but not P2w1 (***Figure 4I***), which was further corroborated by ICS assays (***Figure 4J***). The frequencies of CXCR5 + cells co-producing IL-2 and IFNγ, as well as cells

co-producing IL-21 and IFNγ, indicated that higher numbers of IFNγ+Env-specific cTfh cells were induced with MPLA at P1w1 and P2w1 (*Figure 4K*). Therefore, formulation of gp140 with MPLA induced strong $T_h1$-polarized GC $T_{fh}$ responses, characterized by a higher magnitude of IFNγ+anti Env $T_{fh}$ cells.

## Induction of persistent anti-Env IgG antibodies with HIV-1 Env/MPLA+QS-21

We next sought to determine whether induction of higher frequencies of Env-specific $T_{fh}1$ cells with MPLA could effectively enhance B cell proliferation and differentiation, in turn, promoting the development of Env antibodies with heightened affinity and increased durability. To this end, we performed a wide repertoire of quantitative and functional assays to assess the magnitude, durability, affinity, avidity, reactivity to Group M consensus protein, and effector functionality of vaccine-induced humoral responses to Env. Longitudinal measures of binding antibody concentrations in sera against C.1086 Env showed rapid and equivalent recall in all animal's post P1, with antibodies reaching peak levels after the second protein boost. While antibody kinetics were comparable across vaccine groups, significantly higher antibody magnitude was elicited by MPLA at week 8 post second boost and until 30 weeks after final immunization (*Figure 5A*). At this protracted memory time point, serum antibody concentrations were, on average, 2.5-fold higher with MPLA (median [ng/ml]; MPLA, 5566 versus CAF01, 2155) (*Figure 5B*). To assess CD4 correlates of antibody durability, we asked whether the magnitude of vaccine-elicited GC $T_{fh}$ subsets after each boost predicted antibody concentrations at week 30. Frequencies of GC $T_{fh}$ cells (at P1w2) were positively associated with antibodies at week 30 (*r*=0.6, p<0.05, data not shown). Two additional key cellular determinants of antibody durability emerged from our analysis: GC $T_{fh}1$ cell magnitude (p<0.05, *r*=0.65) (*Figure 5C*) and frequencies of IFNγ+IL-2+Env-specific LN $T_{fh}$ cells (p<0.05, *r*=0.62; *Figure 5D*). These data in concert with IFNγ-regulated molecular programs induced in LN CD4 T cells support a role for GC $T_{fh}1$ induction in antibody persistence.

Using a binding antibody multiplex assay (BAMA), we further confirmed that stronger memory responses were elicited by MPLA, evidenced by higher median serum IgG antibody concentrations to C.1086 gp140 and gp120 antigens measured as area-under-the curve (AUC) over time (*Figure 5E–F*). Thus, utilizing two orthogonal approaches, the data showed significantly higher and protracted serum antibody responses against autologous Env with MPLA, indicative of efficient $T_{fh}$ cell mediated B cell recruitment into the long-lived plasma cell pool.

We next sought to understand if induction of a $T_{fh}1$ profile was also associated with improved Env reactivity to Group M consensus proteins. Therefore, we tested serum reactivity against an antigen panel comprising of consensus gp140 proteins and found that antibodies with higher cross-clade reactivity were induced with MPLA relative to CAF01 (*Figure 5G–H*). We extended these analyses to assess reactivity to V1V2 loops of 1086 C and noted that antibodies with significantly higher specificity to these important regions within Env were induced in both vaccine regimens (*Figure 5I*).

Having established induction of robust serum Env antibodies with MPLA, we next determined whether Env-specific IgG reactivity would also be higher within rectal secretions. After normalization for total IgG levels within rectal secretions, the data showed that Env antibodies were effectively recalled in most animals at week 2 post second protein boost and persisted in all animals 8 weeks after final immunization. However, akin to the systemic compartment, gp140 IgG in rectal secretions were significantly higher with MPLA at memory time point (*Figure 5J*), with sixfold higher levels in MPLA relative to CAF01 at week 8 after final immunization (*Figure 5K*). In both vaccine regimens, the induction of Env-IgA antibodies was poor, preventing a quantitative assessment of IgA responses in secretions.

## Induction of IgG1 subclass antibodies with greater effector functions with HIV-1 Env/MPLA+QS-21

Since antibody effector functions, facilitated by interaction of Ig constant region with cognate Fc receptors on innate cells, mediate protection from acquisition and viral control (*Carpenter and Ackerman, 2020*), we assessed the IgG subclass profile at week 2 post second protein boost. Both CAF01 and MPLA +QS-21 induced Env-specific IgG1, and modest levels of IgG2, and IgG4 isotypes with higher IgG1 in all animals, with exception of one animal with higher IgG4 (*Figure 6—figure supplement 1*).

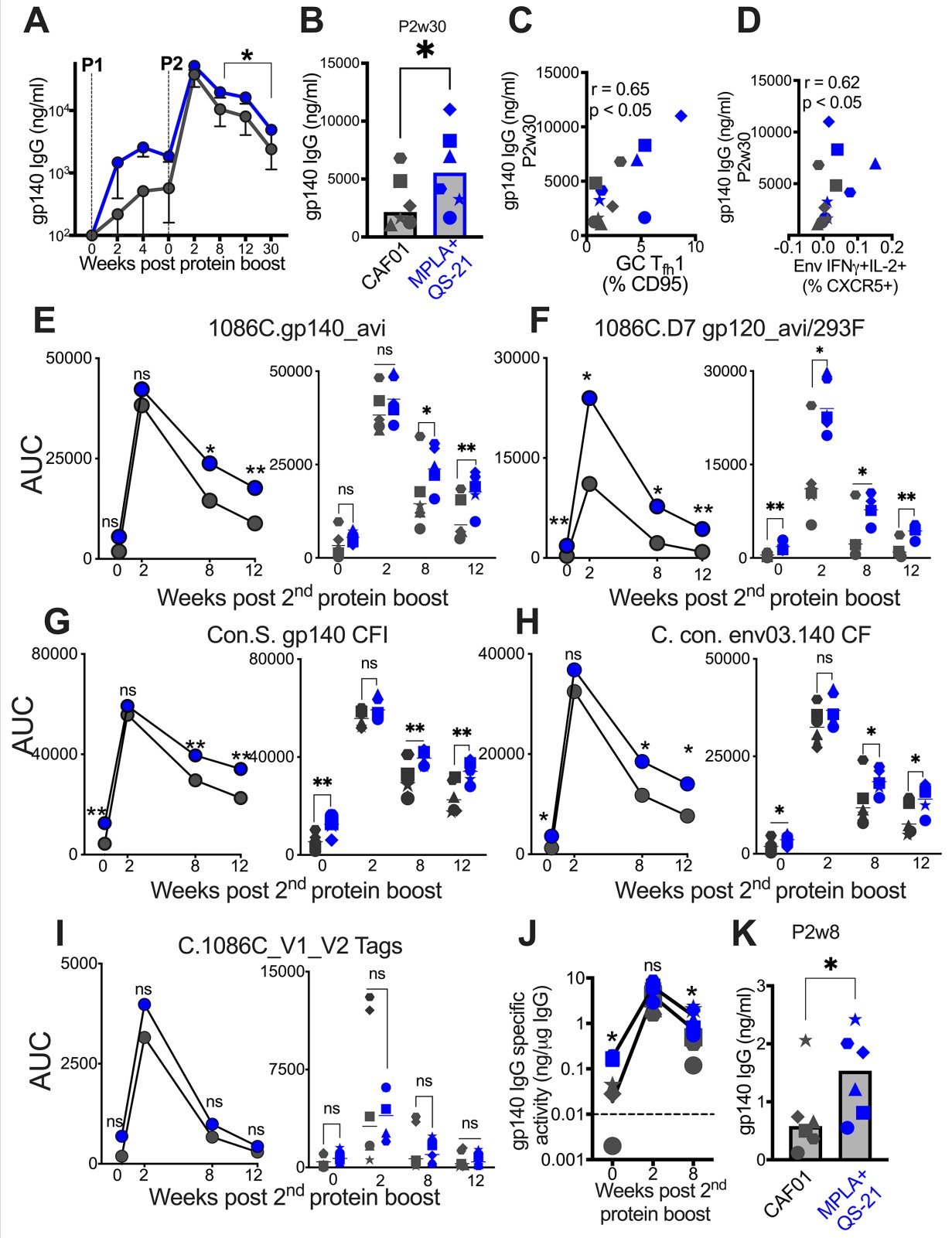

**Figure 5.** Induction of persistent anti-Env IgG antibodies with HIV-1 Env/MPLA +QS-21. (**A**) Kinetics of HIV-1 Env IgG post protein boost (P1 and P2) across vaccine regimens measured by ELISA against C. 1086 gp140. (**B**) Serum gp140 IgG week 30 post P2w3. (**C**) GC $T_{fh}1$ cells correlate with gp140 IgG at week 30. (**D**) Env-specific (IFN-g+IL-2+) $T_{fh}$ cells correlate with gp140 IgG at week 30. (**E–I**) Area under the curve (AUC) values of IgG temporally shown by antigens, as indicated. (**J**) Temporal and (**K**) week 8 measures of gp140-specific IgG levels relative to total IgG in rectal secretions. Data points

*Figure 5 continued on next page*

*Figure 5 continued*

show individual animals. Statistical analysis was performed using Mann-Whitney U test (in panels **A-B**, **E–K**), or two-tailed Spearman rank correlation test (**C–D**); * p<0.05, **p<0.01.

Significantly higher IgG1 was induced by MPLA relative to CAF01 (*Figure 6A*), while IgG2 and IgG4 antibody subclasses were comparably induced (*Figure 6B–C*). The improved IgG1 subtype antibody responses with MPLA prompted us to determine whether a corresponding increase in antibody effector functions might also be observed. Measurement of antibody-dependent phagocytosis (ADP) suggested enhanced effector functions with MPLA (*Figure 6D–E*) which strongly correlated with Env IgG1 but not IgG2 or IgG4 (*Figure 6F*).

Finally, we observed higher levels of neutralizing antibodies against the tier 1A MW965.26 at peak (2 weeks post second protein boost) with MPLA relative to the CAF01 platform (*Figure 6G*). Antibody avidity measurements revealed relatively higher avidity with MPLA against autologous Env and to consensus Env proteins but not V1V2 loops (*Figure 6H–L*). Collectively, these findings provide evidence for significantly improved antibody quality associated with induction of GC $T_{fh}1$ cells.

## Differentiation of GC $T_{fh}$ subsets initiated during the DNA priming phase

Finally, we sought to determine the degree to which $T_{fh}$ responses elicited during the priming phase influenced antibody responses post first protein boost. To this end, we analyzed blood samples at

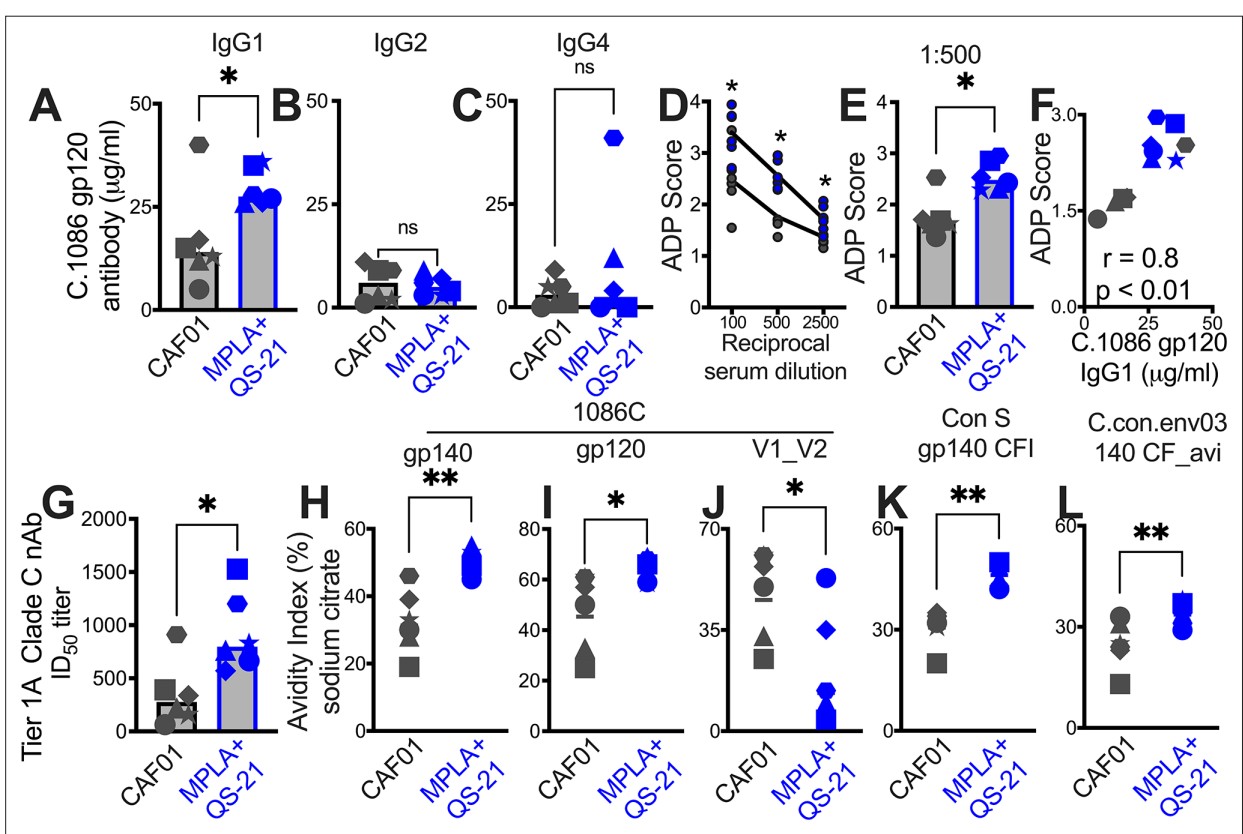

**Figure 6.** Induction of IgG1 subclass antibodies with greater effector functions with HIV-1 Env/MPLA +QS-21. Serum C. 1086 gp120-specific (**A**) IgG1, (**B**) IgG2 (**C**) IgG4 antibodies at P2w2. (**D–E**) Antibody-dependent phagocytosis (ADP) score at P2w8. (**F**) C. 1086 gp120-specific IgG1 correlates with ADP score. (**G**) Infectious dose 50% (ID50) titers to Tier 1 A Clade C MW965.26 HIV-1 isolate at P2w2. (**H–L**) Avidity index (with sodium citrate) across vaccine regimens against specific antigens at P2w8. Data points show individual animals. Statistical analysis was performed using two-tailed Mann-Whitney U test (in panels **A-E**; **G–L**), or two-tailed Spearman rank correlation test (**F**); * p<0.05, **p<0.01.

The online version of this article includes the following figure supplement(s) for figure 6:

**Figure supplement 1.** Serum C. 1086 gp120-specific IgG1 (grey), IgG2 (pink), and IgG4 (black) in animals (x-axis – animal IDs) P2w3.

weeks 0, 1, 2, and 4, while also evaluating FNA of the draining LN at week 2 after the 3rd DNA prime (**Figure 7A**).

Animals were immunized with C.1086 DNA expressing IP10 (DNA$_{IP10}$) which primes stronger GC T$_{fh}$ responses relative to DNA-alone (**Verma et al., 2019**). To further enhance GC T$_{fh}$ responses, beyond that induced by DNA$_{IP10}$, we engineered DNA$_{IP10}$ to co-express IL-6, a key cytokine promoting T$_{fh}$ differentiation. Our vector achieved coordinate expression of Gag and Env antigens together with IP10 +IL6 as confirmed in transfected 293T cells (**Figure 7B**). Following three rounds of intradermal immunization with DNA$_{IP10}$ or DNA$_{IP10+IL6}$, we assessed proliferating cTfh cells (Ki-67$^+$ PD-1$^+$ of CXCR5 +CD4 T cells in blood, **Figure 7C**). We observed a significant increase at weeks 1 and 2 with DNA$_{IP10+IL6}$ but not DNA$_{IP10}$ indicating that IP10 +IL6 incorporation promoted T$_{fh}$ differentiation. Additionally, T$_{fh}$ induction was accompanied by notable T$_{fh}$1 skewing within proliferating cTfh cells (median cTfh1 at day 0 (% CD95), 0.89% vs 1.48% at day 7 and 14). Measurement of antigen-specific responses by AIM assay demonstrated induction of Env and Gag-specific cTfh cells with a trend for higher responses elicited by DNA$_{IP10+IL6}$ (median, 0.42%, IQR 0.3–1.7%) relative to D$_{IP10}$ (median, 0.12%; IQR: 0.04–1.4%, **Figure 7—figure supplement 1**).

Based on the transient accumulation of cTfh cells in the blood, we further explored GC T$_{fh}$ responses. Out of 12 collected FNAs, 8 were successful (four per vaccine group), and demonstrated significant increase in GC T$_{fh}$ frequencies following vaccination (**Figure 7D**). Gating on Ki-67 + cells encompassing both CXCR5 + PD-1+and PD-1 ++ subsets, an increase in activated T$_{fh}$ cells within the LN was observed (**Figure 7E**), accompanied by a corresponding increase in the T$_{fh}$1 subset (**Figure 7F**). Altogether, these data demonstrate induction of GC T$_{fh}$1 responses during the priming phase post immunization with DNA$_{IP10}$ or DNA$_{IP10+IL6}$.

Measurement of C.1086 Env antibodies in serum revealed induction of antibody responses after the 3rd prime in 8/12 animals, with comparable kinetics across vaccine groups. Notably, antibody concentrations 4 weeks post 3rd DNA prime strongly correlated with both the frequencies of proliferating T$_{fh}$ cells (Ki-67 +PD-1+CXCR5+) within the LN and with the proliferating T$_{fh}$1 (Ki-67 +PD-1+CXCR5+CXCR3+) subset at 2 weeks post 3rd DNA prime (**Figure 7G–H**). Moreover, the strong association of antibody concentrations after the 3rd DNA prime with peak antibodies following the first protein boost (**Figure 7I**) indicated efficient recall of Env-specific memory B cells induced during the prime. Overall, the data show that stimulation of T$_{fh}$1 cells during the DNA priming stage is closely linked with robust antibodies post boost.

In summary, our results strongly support maximizing GC T$_{fh}$1 responses during both the prime and boost phases as a strategy to generate potent and durable humoral immunity. These findings have significant implications for the development of more effective vaccination strategies aimed at eliciting robust and long-lasting immune responses.

## Discussion

T$_{fh}$ cells are fundamental in establishing long-lasting humoral immunity with vaccination. The dynamic interplay between innate immune activation and the ensuing inflammatory response directs the magnitude and composition of T$_{fh}$ cell populations, ultimately shaping quantitative and qualitative features of humoral memory (**Hirota et al., 2013**; **Barbet et al., 2018**; **Olatunde et al., 2021**). Here we show that maximizing induction of CXCR3 +T$_{fh}$1 cells correlates with serum Env antibodies which exhibit higher persistence. Our studies with LAMV revealed the robust induction of cTfh1 and GC T$_{fh}$1 responses, along with a moderate induction of T$_{fh}$17 and T$_{fh}$1/17 cells. These findings led us to compare the adjuvants MPLA +QS-21 and CAF01, known to elicit T$_h$1 and mixed T$_h$1/17 responses (**Christensen et al., 2019**), (**Pedersen et al., 2018**) respectively, to determine the optimal strategy for maximizing Env antibody persistence. Through proteomic and transcriptomic analysis of lymph nodes, we uncovered evidence of robust GC responses induced by both vaccine adjuvants following a DNA prime. Moreover, we observed notable skewing of T$_{fh}$ responses, which corresponded with the elicitation of distinct immunogenicity profiles. Interesting, although peak antibody responses following each protein boost were similar across the vaccine platforms, our investigation revealed a notable distinction in antibody persistence. Specifically, the MPLA +QS-21 regimen exhibited higher antibody levels at week 8, 12, and 30, suggestive of enhanced support from T$_{fh}$ cells in promoting plasma cell differentiation. Moreover, T$_{fh}$1 cell induction during the DNA priming phase was linked to effective boosting upon subsequent protein immunization. These findings highlight the critical role of T$_{fh}$1 cells

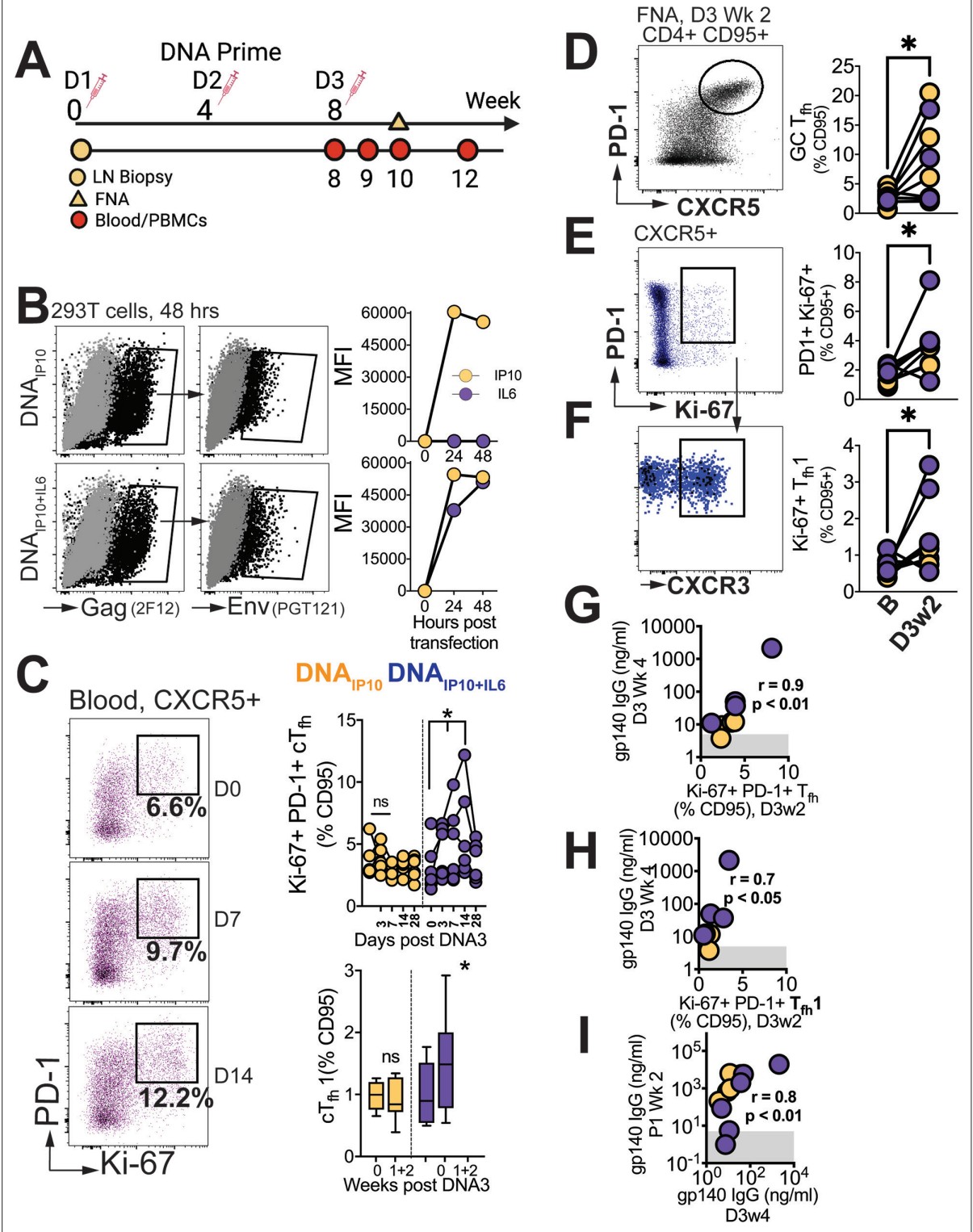

**Figure 7.** Differentiation of GC T$_{fh}$ Subsets initiated during the DNA priming phase. (**A**) Experimental design of DNA immunization phase; FNA, lymph node fine needle aspirates. (**B**) Intracellular expression of Gag (2F12), surface expression of HIV-1 Env (PGT121) and IP10 and IL6 in supernatants of transfected 293T cells. (**C**) Flow cytometry plots show activated (PD-1+Ki-67+) cTfh cells, frequencies post DNA3 (right top); frequencies of cTfh1 cells (right bottom). (**D**) GC T$_{fh}$ cells, (**E**) Ki-67+ PD1+ cells of CXCR5 +subset in lymph node. (**F**) CXCR3 +subset of Ki-67+ PD-1+ CXCR5+ subset. (**G**) Ki-67 +PD-1+T$_{fh}$ in LN correlate with gp140 IgG at week 4 post DNA3. (**H**) Ki-67 +PD-1+T$_{fh}$1 in LN correlate with gp140 IgG at week 4 post DNA. (**I**) gp140 IgG at

*Figure 7 continued on next page*

*Figure 7 continued*

week 2 post P1 correlates with gp140 IgG at week 4 post DNA3. Data points show individual animals. Statistical analysis was performed using one-tailed Wilcoxon matched-pairs signed rank test (in panels C-F), or Spearman rank correlation test (**G–I**); * p<0.05.

The online version of this article includes the following figure supplement(s) for figure 7:

**Figure supplement 1.** Frequencies of OX40$^+$CD40L$^+$ cTfh cells stimulated ex vivo with Gag and Env.

in driving the longevity of antibody responses and emphasize the potential of stimulating T$_{fh}$1 cells during the prime and boost to elicit durable humoral immunity against HIV Env.

A significant finding in RV144 vaccine recipients was the correlation between Env-specific circulating CD4 T cell subsets co-producing IFNγ, TNFα, CD40L, and IL4, and a reduced risk of HIV acquisition [40]. This correlation suggests that productive GC responses elicited by the vaccine fostered protective humoral immunity (*Lin et al., 2015*). More recently, follow up analysis of this population in n=6 subjects immunized with the RV144 vaccine regimen in South Africa (HVTN 097) reported a pT$_h$2 biased transcriptional profile based on observed IL4 and IL13 transcripts (*Cohen et al., 2022*). However, it is important to note that in addition to T$_h$2 cells, CD4 T$_{fh}$ cells also produce IL-4 in a SLAM/SAP-dependent manner (*McCausland et al., 2007*) and T$_h$1 cells can produce IL-13 in the presence of IL18 (*Hayashi et al., 2007*). These findings indicate the need for utilizing multiple immune parameters when categorizing CD4 T cell help and underscore the importance of studying both lymph node and peripheral blood CD4 T$_{fh}$ cells to fully understand their respective roles in fostering Env antibody responses.

In the context of vaccination, our observed association between T$_{fh}$1 cells and persistent antibody responses raises the possibly that higher relative production of IL-21 and CD40L may support the proliferation of GC B cells in concert with IFNγ-mediated B cell proliferation. This contention is supported by the observation of higher relative Ki-67 protein levels in the MPLA group at week 2 of the first protein boost. In the context of a T$_h$1 driven GC response rich in CXCL chemokines CXCL9, 10, and 11; CXCR3 expression on T$_{fh}$ cells may furthermore enable proximity and interaction with GC B cells. CAF01 is composed of cationic liposomal vesicle (dimethyldioctadecylammonium, DDA) together with a synthetic analog of a mycobacterial cell wall (trehalose dibehenate, TDB), a potent activator of macrophages and dendritic cells through the macrophage inducible Ca2$^+$-dependent lectin receptor Mincle, a pattern recognition receptor recognizing bacterial lipids (*Davidsen et al., 2005*; *Martínez-López et al., 2019*). CAF01 elicits CD4 T$_h$1 responses while concurrently driving strong T$_h$17 responses to a mycobacterium TB subunit vaccine via induction of proinflammatory cytokines IL-1β, IL-6, and TNF-α (*Werninghaus et al., 2009*). We observed that CAF01 was effective in inducing robust cellular responses, which were skewed towards a T$_h$1/T$_h$17 profile within GCs. It is possible that employing the CAF06 platform which incorporates MPL into the bilayer of DDA/TDB liposomes may drive stronger T$_h$1 responses together with T$_h$17 cells thereby enhancing protective immunity by eliciting strong humoral immune responses.

The present studies are the first to our knowledge to assess immunological responses across HIV vaccine platforms with MPLA and CAF01-based adjuvants and strongly support GC T$_{fh}$1 cell induction for enhancing Env antibody responses. However, our study has several limitations. First, priming modalities across MPLA and CAF01 were not identical. Animals boosted with MPLA were primed with DNA$_{IP10}$, whereas CAF01 boosted animals received the DNA$_{IP10+IL6}$ prime. Given the stronger immunogenicity of DNA$_{IP10+IL6}$, it is possible that our studies underestimate immune recall potential of MPLA relative to CAF01. Second, 5/6 animals in each vaccine group were males and therefore our findings do not capture possible variability in vaccine response between sexes. Because females develop higher antibody responses than males, it is important to test our hypothesis in animals of both sexes (*Fischinger et al., 2019*). Third, the interplay of innate immune cells and the follicular dendritic cell network in regulation of T$_{fh}$ help for antibody responses is an important consideration for future studies. In this context, conducting mechanistic studies involving cytokine blockade to assess its impact on T$_{fh}$ differentiation in vivo is essential for establishing the significance of adjuvants in shaping humoral immunity. Finally, we did not assess vaccine efficacy and therefore whether GC T$_{fh}$1 cells constitute an important correlate of protection from acquisition remains an important unanswered question.

In summary, our findings demonstrate significant skewing of the GC T$_{fh}$ response by adjuvants inducing distinct inflammatory responses. We further demonstrate that induction of T$_{fh}$1 cells results

in superior persistence of Env antibodies indicating that strategies to harness this CD4 subset may promote protective humoral immunity against HIV.

## Materials and methods

### Rhesus macaques

For LAMV studies, sixteen female colony-bred Indian origin rhesus macaques (*Macaca mulatta*) were utilized. Animals were dichotomized by age into two groups; young [n=8, mean = 4 years] and aged [n=8, mean = 16 years]. Young adults with a higher baseline weight [mean = 5.9 kg] were selected for the cohort to closer match the aged animals in size [mean = 8.6 kg]. For the HIV vaccine study, 12 adult (10 males and 2 females) Indian origin rhesus macaques (*Macaca mulatta*) were utilized. At study initiation, animals were 3.4–5.6 years of age with a median weight of 6.8 kg. Animals in both studies were SIV negative (SIV-), simian T-cell leukemia virus negative (STLV-), and simian retrovirus negative (SRV-); and had no history of dietary, pharmacological, or surgical manipulation. All animals were bred and housed at the California National Primate Research Center (CNPRC) in accordance with the American Association for Accreditation of Laboratory Animal Care (AAALAC) guidelines. All studies were approved by the University of California, Davis Institutional Animal Care and Use Committee (IACUC).

### LAMV immunization

The animals received an IM booster vaccination of canine distemper-measles vaccine (Vanguard).

### HIV-1 immunization

Animals were immunized with DNA vaccines intradermally in both thighs at weeks 0, 4, and 8. At each prime, animals received 4 mg of the pGA2/JS2 plasmid DNA encoding either SHIV C.1086 T/F Env +rhesus interferon-induced protein (IP)–10 (Group 1; n=6) or SHIV C.1086 T/F Env +rhesus interferon-induced protein (IP)–10+rhesus interleukin (IL)–6 (Group 2; n=6). Subsequent protein boosters were administered at weeks 12 and 20 with Group 1 animals receiving 50 µg C.ZA 1197 MB gp140 protein adjuvanted with MPLA +QS-21 and Group 2 animals receiving 50 µg C.ZA 1197 MB gp140 protein adjuvanted with CAF01 delivered in a 175 µL volume subcutaneously into each thigh.

### Cationic adjuvant formulation 01 (CAF01)

CAF01 was generously provided by Statens Serum Institut, Denmark. Admixing with protein was performed by addition of 25 µg of C.ZA 1197 MB gp140 protein (Immune Technology, USA) to 250 µL of CAF01 (625 µg DDA +125 µg TDB) adjuvant formulation. The solution was vortexed vigorously for 15–30 s followed by visual inspection. Remaining protein was added in 10 µg increments with intermittent vortexing until a final amount of 100 µg gp140 was achieved.

### Monophosphoryl lipid A (MPLA)

Synthetic MPLA was purchased from InvivoGen, USA. 100 µg C.ZA 1197 MB gp140 protein was dissolved in a solution of 100 µg MPLA with 50 µg QS-21 saponin [Desert King International, USA] (2:2:1) in PBS.

### Specimen collection and processing

Lymph node (LN) biopsies were obtained at baseline (week 0), week 14 (week 2 post first protein boost) and week 23 (week 3 post second protein boost) and processed as described previously (*Verma et al., 2019*). Isolated cells were washed in complete media, counted, and cryopreserved until subsequent analysis. Two weeks after the 3rd DNA immunization, fine needle aspirates of LN were obtained using a 22-gauge needle, as previously described (*Verma et al., 2019*). PBMCs were isolated from whole blood collected in CPT vacutainer tubes at weeks 0, 4, 9, 10, 13, 21, and 32. Serum was collected from animals at weeks 0, 4, 8, 8+day 3, 9,10, 12, 13, 14, 16, 20, 20+day 3, 21, 22, 23, 28, 32, 50 timepoints and stored at –80 °C until subsequent analysis. Rectal secretions were collected for assessment of mucosal antibody production using premoistened Weck-Cel sponges as previously described (*Verma et al., 2019*).

## Activation induced marker and intracellular cytokine staining assay

Detection of antigen specific CD4 T cells detection was quantified by activation induced marker (AIM) and intracellular cytokine staining (ICS). PBMCs/LN cells were stimulated with overlapping peptide pools of HIV consensus C and HIV-1 C.1086 Env gp140C protein (NIH AIDS Reagent Program) in AIM/R10 media in the presence of 0.2 µg CD28/49d co-stimulatory antibodies (BD) per test. As a positive control, cells were stimulated with 1 X Cell Stimulation Cocktail (PMA and ionomycin) (eBioscience, USA). Unstimulated controls were treated with volume-controlled DMSO (Sigma-Aldrich). Tubes were incubated in 5% $CO_2$ at 37 °C overnight for AIM assay. For ICS assay, after 1 hr of stimulations, protein transport inhibitors 2 µl/mL GolgiPlug (Brefeldin A) and 1.3 µl/mL GolgiStop (Monensin) (BD, Biosciences, USA) were added to the tubes for 8 hr at 37 °C, 5% $CO_2$. Following stimulation, the cells were stained for AIM and ICS surface markers (see *Table 1*). Cells were then fixed with cytofix/cytoperm for 10 min at 4 °C, permeabilized with 1 X Perm wash buffer (BD, Biosciences, USA), and stained for intracellular markers (see *Table 1*) for 45 min. Cells were then washed and acquired the same day on a BD FACS Symphony.

## $T_{fh}$ cell staining by flow cytometry and cell sorting

$T_{fh}$ cell staining was performed on whole blood and LN cells as previously described (*Verma et al., 2019*). Samples were acquired on BD FACS Symphony with FACS Diva version 8.0.1 software and data were analyzed using FlowJo (Versions 9 and 10). For cell sorting, cryopreserved cells were enriched by NHP CD4 isolation kit (Miltenyi Biotec,USA) and stained with CD3, CD4, CXCR5, CD95, and live/dead in complete media (incubated for 1 hr at 4 °C on a shaker). Stained cells were washed twice with 5 mL RPMI plain media (Gibco, USA) and resuspended in 0.5 mL of sorting buffer containing 2% FBS +PBS. The cells were then filtered through a 40 µM strainer into 5 mL sterile FACS tube (Blue cap tube). Cell sorting was performed using a BD FACSAria III. Naive, and CD95 +CD4 T cell populations were collected in complete media supplemented with 20% FBS. Sorted cells were stimulated with overlapping peptide pools of HIV consensus C and HIV-1 C.1086 Env gp140C protein (NIH AIDS Reagent Program) in R10 media in the presence of 0.2 µg/mL CD28/49d co-stimulatory antibodies (BD) for 14 hr. The culture supernatant and stimulated cells were collected and stored at –80 °C for subsequent analysis.

## Serum IL-21 ELISA

Serum IL-21 cytokine was quantified using an IL-21 ELISA Development kit (Novus Bio, USA) in accordance with manufacturer's protocol. Briefly, the capture mAb (MT216G) was diluted to 2 µg/mL in PBS, pH 7.4 and added (100 µL/well) to 96-well microtiter plates with high binding capacity (Thermo Fisher, USA) and incubated overnight at 4 °C. The next day, plates were aspirated, and wells blocked with 200 µL/well of PBS with 0.05% Tween 20 and 0.1% BSA (incubation buffer). After 1 hr of incubation, plates were washed five times with PBS containing 0.05% Tween 20 (300µL/well). Samples or working standards were added at 100 µL/well and incubated for 2 hr at room temperature. After washing, plates were incubated for 1 hr with 100 µL/well of detection mAb diluted to 1 µg/mL in incubation buffer. Streptavidin-HRP conjugates were diluted 1: 1000 in incubation buffer and added to the plates at 100 µL/well for 1 hr followed by washing. Plates were washed and then developed with TMB substrate (Thermo Fisher, USA), and the reaction quenched with 0.2 M $H_2SO_4$ (Sigma, USA). Absorbance was recorded using a Spectramax 5 plate reader (Molecular Devices) at 450 nm with a reference filter at 570 nm within 15 min. The concentration of IL-21 in serum was calculated based on an IL-21 standard curve using SoftMax Pro.

## RNA sequencing and bioinformatics

RNA was extracted from sorted naive, and CD95 +CD4 T cells using a RNeasy plus mini kit (QIAGEN) (*Table 2*). Isolated RNA sample quality was assessed using a BioAnalyzer RNA pico assay (Agilent Technologies Inc, California, USA) and quantified by Qubit 2.0 RNA HS assay (Thermo Fisher, Massachusetts, USA). Library construction was performed based on manufacturer's recommendation for the SMART-Seq v4 Ultra Low Input RNA Kit (Takara Bio USA Inc, California, USA) followed by the Nextera XT DNA Library Prep Kit (Illumina, California, USA). Final library quantity was measured using the KAPA SYBR FAST qPCR and library quality evaluated using a TapeStation D1000 ScreenTape (Agilent Technologies, CA, USA). Final library size was about 450 bp with an insert size of about 300 bp. Illumina

**Table 1.** Flow cytometry antibodies.

| Antibody name | Panel | Vendor | Catalog number/Identifier |
|---|---|---|---|
| Mouse anti-human CD3 (Clone SP34-2) | T$_{FH}$/AIM/ICS | BD Biosciences | Cat#557917; RRID: AB_396938 |
| Mouse anti-human CD4 (Clone L200) | T$_{FH}$/AIM/ICS | BD Biosciences | Cat#563737; RRID: AB2687486 |
| Mouse anti-human CD8 (Clone SK-1) | T$_{FH}$/AIM/ICS | BD Biosciences | Cat#564913; RRID: AB_2833078 |
| Mouse anti-human CD14 (Clone MSE2) | T$_{FH}$ Panel | BioLegend | Cat#301822; RRID: AB_493747 |
| Mouse anti-human CD16 (Clone 3G8) | T$_{FH}$ Panel | BD Biosciences | Cat#563172; RRID: AB_2744297 |
| Mouse anti-human CD20 (Clone 2H7) | T$_{FH}$ Panel | BioLegend | Cat#302314; RRID: AB_314262 |
| Mouse anti-human CD69 (Clone FN50) | T$_{FH}$ Panel | BioLegend | Cat#310944; RRID: AB_2566466 |
| Mouse anti-human CD95 (Clone DX2) | T$_{FH}$/AIM/ICS | BioLegend | Cat#564710; RRID: AB_2738907 |
| Mouse anti-human CXCR3 (CD183) (Clone 1C6) | T$_{FH}$ Panel | BD Biosciences | Cat# 550967; RRID: AB_398481 |
| Mouse anti-human CXCR5 (CD185) (Clone MU5UBEE) | T$_{FH}$/AIM/ICS | eBioscience | Cat#12-9185-42; RRID: AB_11219877 |
| Mouse anti-human CCR6 (CD196) (Clone G034E3) | T$_{FH}$ Panel | BioLegend | Cat#353430; RRID: AB_2564233 |
| Armenian Hamster anti- ICOS (CD278) (Clone C396.4A) | T$_{FH}$ Panel | BioLegend | Cat#313534; RRID: AB_2629729 |
| PECy7 anti-human PD1 (CD279) (Clone EH12.2H8) | T$_{FH}$ Panel | BioLegend | Cat# 329918, RRID: AB_2159324 |
| Mouse anti-human Bcl-6 (Clone K112-91) | T$_{FH}$ Panel | BD Biosciences | Cat# 563581 |
| Mouse anti-Ki-67 (Clone B56) | T$_{FH}$ Panel | BD Biosciences | Cat#558616; RRID: AB_10611866 |
| Mouse anti-human CD25 (Clone BC96) | AIM assay | eBioscience | Cat#**53-0259-42** RRID: AB_2043827 |
| Mouse anti-human CD134 (OX-40) (Clone L106) | AIM assay | BD Biosciences | Cat#744746; RRID: AB_2742454 |
| Mouse anti-human CD137 (4-1BB) (Clone 4B4-1) | AIM assay | BioLegend | Cat# 309826; RRID: AB_2566260 |
| Mouse anti-human CD154 (CD40L) (Clone 24–31) | AIM assay | eBioscience | Cat#17154842 RRID:AB_1582215 |
| Mouse anti-human TNF-α (Clone Mab11) | ICS assay | BioLegend | Cat# 502906; RRID: AB_315258 |
| Mouse anti-human IFNγ (Clone B27) | ICS assay | BioLegend | Cat# 506518; RRID: AB_2123321 |
| Mouse anti-human IL2 (Clone MO1-17H12) | ICS assay | BioLegend | Cat# 500344; RRID: AB_2564091 |
| Mouse anti-human IL-17 (Clone eBio64DEC17) | ICS assay | eBioscience | Cat# 48-7179-42; RRID: AB_10853643 |
| Mouse anti-human IL-21 (Clone 3A3-N2.1) | ICS assay | BD Biosciences | Cat# 560493; RRID: AB_1645421 |
| APC-Cy7 live/dead | | Life Technologies | Cat#L34976 |
| BV510 live/dead | | Life Technologies | Cat#L34966 |

**Table 2.** Samples for RNA seq and Spatial profiling.

| | | | Lymph node collection time points and cells used for | | |
|---|---|---|---|---|---|
| S.No | Animals ID | Vaccine group | Baseline (WK0) | Week2 post 1st protein boost | Week3 post 2nd protein boost |
| 1 | 47161 | MPLA +QS-21 | RNA seq | RNA seq | RNA seq |
| 2 | 45781 | MPLA +QS-21 | Spatial profiling | Spatial profiling | Spatial profiling |
| 3 | 46235 | MPLA +QS-21 | Spatial profiling | Spatial profiling | Spatial profiling |
| 4 | 46551 | MPLA +QS-21 | Spatial profiling | Spatial profiling | Spatial profiling |
| 5 | 46548 | MPLA +QS-21 | RNA seq | RNA seq | RNA seq |
| 6 | 47081 | MPLA +QS-21 | RNA seq | RNA seq | RNA seq |
| 7 | 45721 | CAF01 | RNA seq +Spatial profiling | RNA seq +Spatial profiling | RNA seq +Spatial profiling |
| 8 | 47154 | CAF01 | NA | NA | NA |
| 9 | 46410 | CAF01 | Spatial profiling | Spatial profiling | Spatial profiling |
| 10 | 46354 | CAF01 | RNA seq | RNA seq | RNA seq |
| 11 | 47466 | CAF01 | RNA seq | RNA seq | RNA seq |
| 12 | 47387 | CAF01 | Spatial profiling | Spatial profiling | Spatial profiling |

8-nt dual-indices were used. Equimolar pooling of libraries was performed based on QC values and sequenced on an Illumina NovaSeq S4 (Illumina, California, USA) with a read length configuration of 150 PE for 40 M PE reads per sample (20 M in each direction). Reference rhesus macaque genome (*Macaca_mulatta* _GCF_003339765.1_Mmul_10) and gene model annotation files were downloaded directly from the genome website. Index of the reference genome was built using Hisat2 v2.0.5 and paired-end clean reads were aligned to the reference genome using Hisat2 v2.0.5. The mapped reads of each sample were assembled by StringTie (v1.3.3b) using a reference-based approach.

The quality of the raw RNA-seq data was assessed using FastQC and poor-quality ends were trimmed (Trimgalore). High-quality sequences were aligned against the *Macaca mulatta* reference genome using STAR aligner v2.7.9, and counts were generated using featureCount (*Dobin et al., 2013*). A list of differentially expressed genes (DEGs) were generated using DESeq2 based on the negative binomial distribution (*Love et al., 2014*). The resulting DEGs between the groups were defined at cut-off criteria of |log$_2$ fold-change|≥1.5 and p-value <0.05 adjusted using the Benjamini and Hochberg's approach for controlling the false discovery rate (*Liao et al., 2014*). Gene set enrichment analysis (*Subramanian et al., 2005*) was used to assess the statistical enrichment of gene ontologies and pathways, and visualized using Clusterprofiler v4.8.1 (2012). All statistical analyses were performed using R 4.2.0 (*Yu et al., 2012*).

## GeoMx digital spatial profiling (DSP)

Formalin-fixed paraffin-embedded (FFPE) blocks were prepared using lymph node tissue from biopsies at baseline, week 2 post protein1, and week 3 post protein 2 immunization timepoints to analyze the spatial protein profiling using the Nanostring GeoMx Digital Spatial Profiler (DSP). Tissue sections of 5 μm thickness were cut from FFPE blocks and mounted on GeoMx-NGS BOND RX slide as per manufacturer's recommendations. Sections were baked at 60 °C for 30 min, deparaffinized, rehydrated in CitriSolv, ethanol and washed in water. For antigen retrieval, slides were placed in a staining jar containing 1 X Citrate Buffer (pH 6) for 15 min using a preheated pressure cooker (high pressure and high temp). The tissue slides were washed with 1 X TBS-T and blocked with Buffer W for 1 hr at room temperature in a closed humidity chamber. Tissue slides were stained with a cocktail of fluorescent morphological markers SYTO13 (nuclear stain; AF532), CD20 (Clone: L26; AF488), CD3 (Clone: CD3-12; AF594), and Ki-67 (Clone: B56; AF647) at 1:10,000, 1:250, 1:100, and 1:1000 dilutions, respectively. For protein detection, a multiplex cocktail of primary antibodies was used from

core panels; GeoMx immune cell profiling panel, GeoMx IO drug target module, and GeoMx immune activation status module (See *Table 2*). In total, 60 regions of interest (ROIs) were selected within GCs based on co-localization of CD3 with CD20 +Ki-67+GC B cells. Indexing oligos were released from each ROI by exposure to UV light as described (*Merritt et al., 2020*), and 10 µl of liquid from above the ROI was collected by a microcapillary tip and deposited in a 96-well plate.

Indexing oligos from each ROI were PCR amplified using GeoMx Seq Code primers. PCR products were pooled and purified twice with AMPure XP beads (Beckman Coulter, Brea, CA). Library concentration and purity were measured using a high-sensitivity DNA Bioanalyzer chip (Agilent Technologies, Santa Clara, CA). Paired-end sequencing was performed on an Illumina HiSeq 2000 instrument (Illumina, San Diego, CA). After sequencing, fastq files were run through the GeoMx NGS Pipeline where reads were trimmed, merged, and aligned to a list of indexing oligos to identify the source probe. Analysis of filtered normalized gene expression data was performed in R with Bioconductor. We profiled 32 proteins along with core protein in GeoMx Human Protein array. We first calculated the signal-to-noise ratio by dividing raw count values by the geometric mean of the negative IgG probes. The data were normalized using three negative control IgG probes with the GeomxTools package. Normalized expression values were used for downstream analyses. Dimensionality reduction analysis was performed with principal component analysis using the facomineR package (*Lê et al., 2008*). Differential expression analysis was conducted with the mixedModelDE function from the GeomxTools package. We tested for differences between time points and/or vaccine groups using linear mixed effect models that incorporated animal ID as a random effect term to account for non-independence of the multiple ROIs sampled per animal. Differential expression results were visualized in heatmap plots generated using the ComplexHeatmap R package (*Gu et al., 2016*).

## Serum IgG ELISA

Serum IgG titers against HIV-1 C.1086 Env gp140 were determined by ELISA as described previously (*Verma et al., 2019*). In brief, 96-well microtiter plates with high-binding capacity (Thermo Fisher, USA) were coated overnight at 4 °C with C.1086 Env gp140 protein from the NIH AIDS Reagent Program (ARP) diluted in coating buffer. Plates were washed and blocked with nonfat dry milk in PBS for two hours at room temperature and washed five times with 1 X PBS-0.05% tween-20 (PBST). Diluted standard and serum samples were added to plate wells and incubated overnight at 4 °C. Plates were washed, and HRP conjugated goat anti-monkey IgG (Nordic MUbio, Netherlands) was added to plate wells and incubated at room temperature for 1 hr. Plates were then washed and developed with the addition of TMB substrate (Thermo Fisher, USA). Absorbance was recorded at 450 nm with a reference filter at 570 nm using a Spectramax 5 plate reader (Molecular Devices, USA). Baseline sera from each animal served as negative control and optical density (OD) values twofold above baseline were considered positive and extrapolated using in-plate standards to determine anti-Env antibody concentrations.

## IgG subclass antibodies

ELISA was used to measure concentrations of gp120-specific IgG1-4 antibodies. Ten rows of a 96-well Immulon 4 microtiter plate (VWR) were coated overnight at 4 °C with 50 ng C.1086 gp120Δ7 (HIV Reagent Program) per well in PBS, pH 7.2. To generate a standard curve with a known amount of IgG1, IgG2, IgG3 or IgG4 antibody, 2 rows of the plate were coated with 50 ng per well recombinant CD40 (NHP Reagent Resource). The following day, the plate was washed with PBST and blocked with PBST containing 0.1% BSA (ELISA buffer: EB). Serum samples diluted in EB were then added to gp120 wells. Duplicate dilutions of anti-CD40 rhesus IgG1, IgG2, IgG3, or IgG4 antibody (NHP Reagent Resource) were added to the CD40 wells. Following overnight incubation at 4 °C, the plate was washed and treated for 1 hr at 37 °C with 0.5 µg/ml of the appropriate biotinylated monoclonal antibody (all from NHP Reagent Resource) diluted in EB: anti-rhesus IgG1 clone 7H11 (ena), anti-rhesus IgG2 clone 8D11 (dio), anti-rhesus IgG3 clone 6F5 (tria) or anti-rhesus IgG4 clone 7A8 (tessera). Plates were washed, treated with 1/4,000 diluted neutralite avidin peroxidase (SouthernBiotech) for 30 min at room temperature, developed with TMB substrate and quenched with a $H_2SO_4$ stop solution. After recording absorbance at 450 nm, a standard curve constructed from the anti-CD40 IgG subclass antibody was used to interpolate the concentration of anti-gp120 IgG of the same subclass in samples.

## Avidity index binding antibody multiplex assay

A binding antibody multiplex assay avidity index (BAMA-AI) method (*Buchbinder et al., 2017*; *Seaton et al., 2021*) was used to measure the strength of IgG antibody-antigen interactions in serum collected at baseline and 0-, 2-, 8-, and 12 weeks post second protein boost. Baseline sera were diluted at 1:80, and post-immunization sera were diluted at 1:80 and titrated 5-fold for 6 dilutions. Diluted serum samples were incubated with a mixture of magnetic bead sets coupled to one of five HIV-1 antigens C.1086 gp140, C.1086_gp120, C.1086_V1V2, Con C (clade C consensus) gp140, Con S (group M consensus) gp140 for 30 min. Beads were then washed and treated with dissociative sodium citrate buffer (pH 4.0, Teknova) or PBS for 15 min, prior to addition of a goat anti-monkey IgG-biotin secondary detection antibody (4 µg/mL; Rockland) for 30 min followed by phycoerythrin (PE) streptavidin (1:100 dilution; BD Pharmingen) for 30 minutes. Beads were acquired on a Bio-Plex 200 instrument (Bio-Rad), with antibody binding expressed as mean fluorescence intensity (MFI). Positive controls included HIV IgG immunoglobulin (HIVIG) and CH58_4 A IgG (HIV-1 V2-specific) monoclonal antibody titrations. Negative controls included blank (uncoupled) beads, normal human reference serum (Sigma-Aldrich), and pooled seronegative monkey plasma (BioIVT). Titer of HIV-1-specific binding antibodies was reported as AUC, calculated across the titrated PBS-treated wells. Avidity of antibody binding was reported as avidity index (a percentage), defined as MFI in the citrate treated well divided by MFI in PBS-treated well x 100. AI was calculated only if the response in the PBS-treated well was considered positive and in the linear range of the assay. Positivity criteria were as follows: (1) MFI >100 (2) MFI >antigen-specific cut-off (95th percentile of all baseline sample binding per antigen after high baseline exclusion), (3) MFI >threefold of matched baseline sample binding before and after blank bead subtraction. Positivity calls were made at the 1:80 dilution for samples tested in PBS. AI values were confirmed to be within 10% across the sample dilutions, with the lowest dilution meeting this criterion selected for reporting. Samples that did not meet the above-mentioned criteria were reported as indeterminate for AI.

## ADP assay

Serum antibodies were evaluated for ability to enhance phagocytosis of gp120 expressing fluorescent beads by THP-1 monocytes as previously described (*Verma et al., 2019*). Briefly, 1 µm avidin-coated fluorospheres (Invitrogen) were labeled with biotinylated anti-His tag monoclonal antibody (Thermo-Scientific) and used to capture His-tagged clade C gp120 Du151 protein (Immune Technologies). The gp120-expressing beads were then incubated with triplicate fivefold dilutions of heat-inactivated serum samples in V-bottom plates for 1 hr at 37 °C. THP-1 cells ($2x10^4$ per well) were then added and incubated at 37 °C in 5% $CO_2$. After 5 h, cells were washed with $Ca^{+2}/Mg^{+2}$-free DPBS, then treated with 0.12% trypsin-EDTA for 5 min at 37 °C. The cells were washed and resuspended in 1% paraformaldehyde. Fluorescence was measured using a FACSCanto (BD Biosciences) and FlowJo 9 software. Phagocytosis was quantified by multiplying the percentage of fluorescent cells by their median fluorescence intensity. A phagocytic score was then calculated by dividing the phagocytosis measured in each test sample by the phagocytosis observed for pooled pre-immune serum at the same dilution.

## Neutralization assay

Serum neutralizing activity was quantified using a previously described pseudovirus infectivity assay where TZM-bl cells were co-incubated with tier 1 clade C pseudovirus MW965.26, tier 2 clade C virus Ce1086_B2, tier 2 clade C Ce1176_A3, or a control MLV-pseudotyped virus (to measure non-HIV-specific activity). Neutralization titers were measured in serum samples collected at week 0 and week 2 following the second protein boost. Samples neutralizing antibody activity against SIV pseudoviruses was measured relative to MLV pseudovirus negative controls.

## Mucosal antibodies

A custom binding antibody multiplex assay (BAMA) assay with C.1086 gp140 K160N-conjugated Bio-Rad Bio-plex Pro magnetic COOH beads was used to quantify Env-specific IgG in rectal secretions and specific IgA in IgG-depleted sera and rectal secretions as previously described (*Verma et al., 2019*). Briefly, 50 µg gp140 was conjugated to $10x10^6$ beads as described (*Iyer et al., 2015*). The gp140-labeled beads (3500 per well) were shaken overnight at 1100 rpm and 4 °C with 10-fold dilutions of serum or secretion and standard *Verma et al., 2019*. Beads were then washed with PBST and reacted

for 30 minutes at room temperature with biotinylated goat anti-human IgG (Southern Biotech) or goat whole IgG containing anti-monkey IgA (Novus) followed by phycoerythrin-labeled neutralite avidin (Southern Biotech). Fluorescence was measured with a Bio-Rad Bioplex 200. Antibody concentrations were determined using standard curves prepared with Bioplex Manager software (Bio-Rad). Concentrations of gp120-specific IgG in secretions were divided by the concentration of total IgG or IgA measured in the sample by ELISA to obtain the specific activity (ng of gp140-specific IgG or IgA antibody per μg of total IgG or IgA, respectively).

## Statistical analysis

Statistical analysis was performed using Prism v.9.5.1 (Graph Pad). Gaussian distribution was not assumed, thus comparisons across time, between groups, and correlations were analyzed using non-parametric tests. For within group comparisons, Wilcoxon matched pairs signed rank test was used. For between group comparisons, Mann-Whitney rank comparisons were used. Correlation coefficients were computed using Spearman correlation. p-values <0.05 were considered significant.

## Acknowledgements

The authors are grateful to Dennis Christensen, previously at the SSI, for provision of CAF01 adjuvant for the studies. The authors are grateful to Brian Schmidt for technical assistance with the LAMV study; Wilhelm Von Morgenland and Miles Christensen and the CNPRC SAIDS team for coordination of macaque studies; Stephenie Liu at the UC Davis Genomics Resource for optimization of morphological markers for spatial profiling, and Daniel Newhouse at the Spatial Data Analysis Service (sDAS) for analysis of spatial proteomics data. We thank Robert L Wilson for excellent technical assistance in processing secretions and measuring mucosal antibodies. The authors thank Aloki Patel for technical assistance with BAMA-AI assays, Kelly Seaton and Angelina Sharak for BAMA-AI data selection/consultation, and Lu Zhang for data analysis and visualization (Center for Human Systems Immunology, Duke University). The authors acknowledge funding support from NIAID, NIH R56AI150409 (SSI) and contract # HHSN272201800004C (XS); Duke CFAR Grant P30 AI064518 (GDT). Reagents for measurement of rhesus IgG subclasses were obtained from the NIH Nonhuman Primate Reagent Resource supported by AI126683 and OD10976.

## Additional information

### Funding

| Funder | Grant reference number | Author |
| --- | --- | --- |
| NIH Office of the Director | K01OD023034 | Smita S Iyer |
| National Institutes of Health | R56AI150409 | Smita S Iyer |
| National Institutes of Health | HHSN272201800004C | Xiaoying Shen |
| National Institutes of Health | P30 AI064518 | Georgia D Tomaras |

The funders had no role in study design, data collection and interpretation, or the decision to submit the work for publication.

### Author contributions

Anil Verma, Data curation, Formal analysis, Investigation, Writing – original draft, Writing – review and editing; Chase E Hawes, Formal analysis, Investigation, Writing – original draft, Writing – review and editing; Sonny R Elizaldi, Justin C Smith, Xiaoying Shen, LaTonya D Williams, Formal analysis, Investigation, Writing – review and editing; Dhivyaa Rajasundaram, Data curation, Formal analysis, Writing – review and editing; Gabriel Kristian Pedersen, Rama R Amara, Resources, Writing – review and editing; Georgia D Tomaras, Investigation, Writing – review and editing; Pamela A Kozlowski, Data curation, Formal analysis, Investigation, Methodology, Writing – review and editing; Smita S Iyer,

Conceptualization, Data curation, Formal analysis, Supervision, Funding acquisition, Investigation, Methodology, Writing – original draft, Project administration, Writing – review and editing

### Author ORCIDs
Justin C Smith ⓘ https://orcid.org/0000-0001-7702-7920
Smita S Iyer ⓘ https://orcid.org/0000-0002-8918-7005

### Ethics
All animals were bred and housed at the California National Primate Research Center (CNPRC) in accordance with the American Association for Accreditation of Laboratory Animal Care (AAALAC) guidelines. All studies were approved by the University of California, Davis Institutional Animal Care and Use Committee (IACUC).

Reviewer #1 (Public Review): https://doi.org/10.7554/eLife.89395.3.sa1
Reviewer #2 (Public Review): https://doi.org/10.7554/eLife.89395.3.sa2
Author Response https://doi.org/10.7554/eLife.89395.3.sa3

---

## Additional files

### Supplementary files
• MDAR checklist

### Data availability
RNA-seq dataset is accessible at GSE234813.

The following dataset was generated:

| Author(s) | Year | Dataset title | Dataset URL | Database and Identifier |
| --- | --- | --- | --- | --- |
| Iyer SS, Verma A | 2023 | CD4 T Follicular Helper 1 Cells Promote HIV-1 Env Antibody Persistence | https://www.ncbi.nlm.nih.gov/geo/query/acc.cgi?acc=GSE234813 | NCBI Gene Expression Omnibus, GSE234813 |

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
