## [Editor Report · eLife assessment]

The authors' findings have theoretical or practical deep implications, which makes them **important**. The methods, data, and analyzes support the authors' arguments with only minor weaknesses, and overall they are **solid**. In vitro culture experiments could provide evidence to strengthen the evidence for the functional significance of Th1-mediated cytokines in the observed B cell responses.

---

## [Referee Report · Reviewer #1 (Public Review)]

Summary

Developing vaccination capable of inducing persistent antibody responses capable of broadly neutralizing HIV strains is of high importance. However, our ability to design vaccines to achieve this is limited by our relative lack of understanding of the role of T-follicular helper (Tfh) subtypes in the responses. In this report Verma et al investigate the effects of different prime and boost vaccination strategies to induce skewed Tfh responses and its relationship to antibody levels. They initially find that live-attenuated measles vaccine, known to be effective at inducing prolonged antibody responses has a significant minority of germinal center Tfh (GC-Tfh) with a Th1 phenotype (GC-Tfh1) and then explore whether a prime and boost vaccination strategy designed to induce GC-Tfh1 is effective in the context of anti-HIV vaccination. They demonstrate that a vaccine formulation referred to as MPLA induces higher GC-Tfh1 and link this to increased antibody production.

Strengths:

While there is a lot of literature on Tfh subtypes in blood, how this related to the germinal centers is not always clear. The strength of this paper is that they use a relevant model to allow some longitudinal insight into the detailed events of the germinal center Tfh (GC-Tfh) compartment across time and how this related to antibody production.

Weaknesses:

The authors focus strongly on the proportion of GC-Tfh1 of GC-Tfh. There seems to be an assumption that since the MPLA vaccine has a higher number of GC-Tfh1 that this explains the higher levels of antibodies. This is not an entirely unreasonable assumption but the mechanistic link between the two is never tested.

---

## [Referee Report · Reviewer #2 (Public Review)]

Summary:

Anil Verma et al. have performed prime-boost HIV vaccination to enhance HIV-1 Env antibodies in the rhesus macaques model. The authors used two different adjuvants, a cationic liposome-based adjuvant (CAF01) and a monophosphoryl lipid A (MPLA)+QS-21 adjuvant. They demonstrated that these two adjuvants promote different transcriptomes in the GC-TFH subsets. The MPLA+QS-21 adjuvant induces abundant GC TFH1 cells expressing CXCR3 at first priming, while the CAF01 adjuvant predominantly induced GC TFH1/17 cells co-expressing CXCR3 and CCR6. Both adjuvants initiate comparable Env antibody responses. However, MPLA+QS-21 shows more significant IgG1 antibodies binding to gp140 even after 30 weeks.

The enhancement of memory responses by MPLA+QS-21 consistently associates with the emergence of GC TFH1 cells that preferentially produce IFN-γ.

Strengths:

The strength of this manuscript is that all experiments have been done in the rhesus macaques model with great care. This manuscript beautifully indicated that MPLA+QS-21 would be a promising adjuvant to induce the memory B cell response in the HIV vaccine.

Weaknesses:

The authors did not provide clear evidence to indicate the functional relevance of GC TFH1 in IgG1 class-switch and B cell memory responses.

---

## [Author Response]

The following is the authors’ response to the original reviews.

eLife assessmentDespite the importance of T follicular helper cells (Tfh cells) in vaccine-induced humoral responses, it is still unclear which type of Tfh cells (Tfh1, Tfh2, and Tfh17) is critical for generating protective humoral immunity. By using the rhesus macaques model (most similar to human), the authors have addressed this potentially important question and obtained suggestive data that Tfh1 is critical. Although being suggestive, the evidence for the importance of Tfh1 is incomplete.
**Public Reviews:**

**Reviewer #1 (Public Review):**
Summary:Developing vaccination capable of inducing persistent antibody responses capable of broadly neutralizing HIV strains is of high importance. However, our ability to design vaccines to achieve this is limited by our relative lack of understanding of the role of T-follicular helper (Tfh) subtypes in the responses. In this report Verma et al investigate the effects of different prime and boost vaccination strategies to induce skewed Tfh responses and its relationship to antibody levels. They initially find that live-attenuated measles vaccine, known to be effective at inducing prolonged antibody responses has a significant minority of germinal center Tfh (GC-Tfh) with a Th1 phenotype (GC-Tfh1) and then explore whether a prime and boost vaccination strategy designed to induce GC-Tfh1 is effective in the context of anti-HIV vaccination. They conclude that a vaccine formulation referred to as MPLA before concluding that this is the case.

Clarification: MPLA serves as the adjuvant, and the vaccine formulation is characterized as a Th1 formulation based on the properties of the adjuvant.

Strengths:While there is a lot of literature on Tfh subtypes in blood, how this relates to the germinal centers is not always clear. The strength of this paper is that they use a relevant model to allow some longitudinal insight into the detailed events of the germinal center Tfh (GC-Tfh) compartment across time and how this related to antibody production.Weaknesses:The authors focus strongly on the numbers of GC-Tfh1 as a proportion of memory cells and their comparison to GC-Tfh17. There seems to be little consideration of the large proportion of GC-Tfh which express neither CCR6 and CXCR3 and currently no clear reasoning for excluding the majority of GC-Tfh from most analysis. There seems to be an assumption that since the MPLA vaccine has a higher number of GC-Tfh1 that this explains the higher levels of antibodies. There is not sufficient information to make it clear if the primary difference in vaccine efficacy is due to a greater proportion of GC-Tfh1 or an overall increase in GC-Tfh of which the percentage of GC-Tfh1 is relatively fixed.

Response: We appreciate the reviewer's comment. Indeed, while there is substantial literature on Tfh subtypes in blood; the strength of our study lies in utilizing a relevant model to provide longitudinal insights into the dynamics of the germinal center Tfh (GC-Tfh) compartment over time and its relationship to antibody production. Regarding the concern about the comprehensive analysis of GC Tfh subsets, including GC-Tfh1, GC-Tfh17, and others not expressing CCR6 and/or CXCR3, we fully acknowledge its importance. To address this, we will conduct a detailed analysis of GC Tfh and GC Tfh1 frequencies, encompassing subsets without CCR6 and CXCR3 expression, to provide a more comprehensive view of the GC-Tfh population in our analysis.

**Reviewer #2 (Public Review):**
Summary:Anil Verma et al. have performed prime-boost HIV vaccination to enhance HIV-1 Env antibodies in the rhesus macaque model. The authors used two different adjuvants, a cationic liposome-based adjuvant (CAF01) and a monophosphoryl lipid A (MPLA)+QS-21 adjuvant. They demonstrated that these two adjuvants promote different transcriptomes in the GC-TFH subsets. The MPLA+QS-21 adjuvant induces abundant GC TFH1 cells expressing CXCR3 at first priming, while the CAF01 adjuvant predominantly induced GC TFH1/17 cells co-expressing CXCR3 and CCR6. Both adjuvants initiate comparable Env antibody responses. However, MPLA+QS-21 shows more significant IgG1 antibodies binding to gp140 even after 30 weeks.The enhancement of memory responses by MPLA+QS-21 consistently associates with the emergence of GC TFH1 cells that preferentially produce IFN-γ.Strengths:The strength of this manuscript is that all experiments have been done in the rhesus macaque model with great care. This manuscript beautifully indicated that MPLA+QS-21 would be a promising adjuvant to induce the memory B cell response in the HIV vaccine.Weaknesses:The authors did not provide clear evidence to indicate the functional relevance of GC TFH1 in IgG1 class-switch and B cell memory responses.

Response. We appreciate the recognition of our meticulous work in the rhesus macaque model and the potential of MPLA+QS-21 as an adjuvant for HIV vaccine-induced humoral immunity. We acknowledge the need to provide clearer evidence of the functional relevance of GC Tfh1 in IgG1 class-switching and B cell memory responses. We will attempt to address this concern in our revisions.

**Recommendations for Authors:**

**Reviewer #1:**
1. Is the proportion of GC-Tfh1 within GC-Tfh significantly increased in MPLA vs CAF01? Thebalance between Tfh1 and Tfh17 data is shown in 4C but appears quite a modest difference.Additionally, it excludes the majority of GC-Tfh since it only considers CCR6 and CXCR3expressing cells.

Response. We have now included a comparison of the relative proportions of GC Tfh cells expressing CCR6 and CXCR3, as well as those lacking these markers. Our data now demonstrate an increased presence of Tfh1 within the GC-Tfh population when MPLA is employed at P1w2, as depicted in Figure 4D.

1. Is there any relationship between GC-Tfh17, 1/17 and non Th1/17 GC-Tfh and antibodylevels? In Figure 5C only GC Tfh1 is examined making it impossible to judge if this is specific toGC-Tfh1 or a general relationship between higher total GC-Tfh and antibodies.

Response. In our revised description of the results, we have mentioned that GC Tfh frequencies correlated with antibody levels (r = 0.6, p < 0.05). However, it is important to note that this correlation was specific to the GC Tfh1 subset and was not observed with other subsets.

Other points:1. The authors make a number of statements that rather exaggerate differences such as stating in the abstract that CAF01 induces Tfh1/17 while MPLA predominantly induces Tfh1. As shown in Figure 4C the majority of CCR6-CXCR3- GC-Tfh induced by CAF01 are GC-Tfh1 i.e. both formulations predominantly induce GC-Tfh1. Also, it is difficult to judge since the data is never provided but the predominant group of GC-Tfh appears to be CCR6-CXCR3- in both cases.

Response. We acknowledge the need for greater precision in our descriptions. In response, we have addressed this concern by providing the frequencies of CCR6-CXCR3- GC Tfh cells in Figure 4D. We have also included a comparison of the relative frequencies across the adjuvant groups in the Results section (Lines 331-338).

1. The authors use the term peripheral Tfh (pTfh), it may be better to use the more common term circulating Tfh (cTfh) to avoid confusion with T peripheral helper cells (Tph).

Response. We appreciate the reviewer's suggestion to use the more commonly accepted term "circulating Tfh (cTfh)" instead of "peripheral Tfh (pTfh)." We have incorporated this change into our manuscript to ensure clarity and avoid potential confusion with "peripheral helper cells (Tph).

1. Some further labelling of the pie chart in Figure 1G to at least specify larger groups such as Tfh2, Tfh17, Tfh1/17 would be helpful.

Response. We have incorporated the suggestion and identified cTfh2, cTfh17, and cTfh2/17 cells. We additionally now state in the legend that overlapping pie arcs correspond to specific polarized Tfh subsets denoted by arc color.

1. A gating example of the CXCR3, CCR6, CCR4 patterns in the GC Tfh would be helpful. "up to 25% of GC Tfh cells expressed CCR6" I think it is better to state the average here since 25% appears an outlier.

Response. We have now included a gating example of chemokine receptor expression, patterns in the GC Tfh. Additionally, we have revised the statement to mention the median (7%) of GC Tfh cells expressing CCR6 instead of specifying the upper limit.

1. Figure 1I, does this graph exclude triple negative cells? It's not clear from the figure legend but the numbers do not seem to add up with the graphical proportions shown in figure 1H.

Response. We have made the necessary clarification in both the results section, figure, and the figure legend to state that the Boolean analysis is based on cells expressing either CXCR3 or CCR6, thus explaining the exclusion of triple negative cells.

1. Figure 3C. Some label should be added to make clear which violins are from the CD95- and CD95+ groups. There may be too much data in this panel for p values to be legible. Either less graphs or more space may be needed.

Response. We have updated the Y axis labels in the figure to state that the violin plots show the differences in gene expression between CD95+ CD4 T cells and CD95- CD4 T cells (naive).

1. Figure 4B. Numbers attached to the gates (1, 17 etc) should be more clearly labeled Tfh1, Tfh17 etc since normally they might be expected to be gate percentages in this format. Gate percentages should also be added.

Response. We have clearly labeled the subsets as "Tfh1" and "Tfh17," making it easier for readers to interpret the figure. Additionally, we have included gate percentages in the flow plot. Furthermore, the percentages of GC Tfh subsets are now depicted in Figure 4D.

1. Overlarge and indistinct datapoint symbols are often a problem e.g. Figure 4G most of the CAF01 datapoints are merged into a single blob with no indication of where one point ends or begins. Supplementary figure 5E. Datapoint sizes are large to the extent that the lines are difficult to see. Lines indicating central tendency are often lost.

Response. We have reworked the graphs (including 4G, now 4I) to ensure clarity,

1. Generally greater care is needed with graph layout e.g. the B indicating figure 6B is on the graph of figure 6A.

Response. We have made the necessary adjustment to ensure that the letter "B" correctly corresponds to the graph in Figure 6B.

1. Figure 6J, the text seems to indicate "higher avidity with MPLA against autologous Env including V1V2 loops." However, the graph seems to indicate lower avidity for V1V2 loops?Response. We appreciate the careful observation. We have rectified this by updating the description in the results section to accurately reflect the graph, which shows higher avidity for V1V2 loops with CAF01.1. Figure 6A. The authors state that significantly higher IgG1 was induced but Figure 6A seems to be the only graph lacking an indication of statistical significance.

Response. We have made the necessary adjustment to ensure that significance symbol is depicted in Figure 6A.

1. Brackets indicating significance are often unclear. e.g. in Figure 4B MPLA graph there are three groups and a single multipoint bracket with a single result making it unclear which groups are being compared.

Response. We have added clarification to the legend. It now states that the temporal comparisons in GC Tfh subsets for each vaccine group are made in relation to frequencies at baseline. This revision provides a clear reference point for the significance comparisons and ensures that readers can easily understand which groups are being compared.

**Reviewer #2:**
Overall, the manuscript is well-written and addresses an important issue. However, further investigation is warranted to understand how the MPLA+QS-21 induced GC TFH1 influenced on memory B cell response. This manuscript only showed the correlation between GC TFH1 and antibody responses. If the authors explain adjuvant preference in memory B cell responses, this manuscript could be more considerable for publication.1. This reviewer recommends that the author provide more evidence to indicate the functional relevance of GC TFH1 in IgG1 class-switch and B cell memory responses. Some evidence supports that IFN-γ controls the antigen-specific IgG1 responses in humans, but it is still controversial. The author also suggests the involvement of IL-21, but this is also an open question even in the human system. This is also the case in the memory responses. There is no direct link between IFN-γ and memory B cell responses in the human system. The authors need more evidence of how GC TFH1 cell development has more advantages in IgG1 and memory responses than GC TFH1 /17 cells. I believe an antibody blockade of cytokines would be a possible strategy to prove these questions.

Response. We appreciate the reviewer's valuable suggestion to provide more evidence regarding the functional relevance of GC Tfh1 cells in IgG1 class-switch and B cell memory responses. It is indeed important to establish a direct link between GC Tfh1 cells and these responses, particularly in the context of cytokine skewing. The suggestion of antibody blockade studies to mechanistically link the modulation of the inflammatory milieu to Tfh differentiation and subsequent antibody functions is important. However, we must acknowledge that these studies are currently beyond the scope of our work. We have included this as a limitation in our study, recognizing the need for further studies to address these important questions.

1. In Fig.5, the authors use different scales to indicate the IgG antibody titer. A shows the log scale, while B shows the linear scale. Moreover, the differences are minimal, even though the authors indicated a significant difference. I am not sure this difference is meaningful.

Response. To clarify, we used a log scale in Figure 5A to demonstrate temporal changes over the course of vaccination. In Figure 5B, where we are comparing differences across vaccine regimens at week 30, a linear scale was deemed more appropriate, as it allows for a clear representation of the approximately two-fold difference observed. We fully acknowledge that to establish the biological significance of the observed difference, challenge studies will be essential.